# Loss of NDR1/2 kinases impairs endomembrane trafficking and autophagy leading to neurodegeneration

Flavia Roşianu[1], Simeon R Mihaylov[1], Noreen Eder[1], Antonie Martiniuc[1], Suzanne Claxton[1], Helen R Flynn[2], Shamsinar Jalal[3], Marie-Charlotte Domart[4], Lucy Collinson[4], Mark Skehel[2], Ambrosius P Snijders[2], Matthias Krause[3], Sharon A Tooze[5], Sila K Ultanir[1]

Autophagy is essential for neuronal development and its deregulation contributes to neurodegenerative diseases. NDR1 and NDR2 are highly conserved kinases, implicated in neuronal development, mitochondrial health and autophagy, but how they affect mammalian brain development in vivo is not known. Using single and double *Ndr1/2* knockout mouse models, we show that only dual loss of *Ndr1/2* in neurons causes neurodegeneration. This phenotype was present when NDR kinases were deleted both during embryonic development, as well as in adult mice. Proteomic and phospho-proteomic comparisons between *Ndr1/2* knockout and control brains revealed novel kinase substrates and indicated that endocytosis is significantly affected in the absence of NDR1/2. We validated the endocytic protein Raph1/Lpd1, as a novel NDR1/2 substrate, and showed that both NDR1/2 and Raph1 are critical for endocytosis and membrane recycling. In NDR1/2 knockout brains, we observed prominent accumulation of transferrin receptor, p62 and ubiquitinated proteins, indicative of a major impairment of protein homeostasis. Furthermore, the levels of LC3-positive autophagosomes were reduced in knockout neurons, implying that reduced autophagy efficiency mediates p62 accumulation and neurotoxicity. Mechanistically, pronounced mislocalisation of the transmembrane autophagy protein ATG9A at the neuronal periphery, impaired axonal ATG9A trafficking and increased ATG9A surface levels further confirm defects in membrane trafficking, and could underlie the impairment in autophagy. We provide novel insight into the roles of NDR1/2 kinases in maintaining neuronal health.

## Introduction

Macroautophagy (hereafter referred to as autophagy) is a degradation process for cytoplasmic organelles and proteins (Ohsumi, 2001; Yorimitsu & Klionsky, 2005; Klionsky et al, 2010; Mizushima & Komatsu, 2011; Yamamoto & Yue, 2014; Yu et al, 2018). Because of its essential role in maintaining neuronal protein and organelle homeostasis, via a constant turnover of these components, constitutive autophagy is indispensable for neuronal survival (Menzies et al, 2017; Azarnia Tehran et al, 2018; Kulkarni et al, 2018; Stavoe & Holzbaur, 2019). Neuronal autophagosomes form predominantly in the axons and presynaptic compartments and are trafficked to the soma, maturing along the way and eventually fusing with fully competent lysosomes in the cell body (Maday et al, 2014; Murdoch et al, 2016; Soukup et al, 2016; Stavoe & Holzbaur, 2019). Impairment of autophagy in brain-specific *Atg7* or *Atg5* knockout mice leads to neurodegeneration (Hara et al, 2006; Komatsu et al, 2006), and alterations in autophagy may also underlie human neurodegenerative diseases (Harris & Rubinsztein, 2011; Wong & Holzbaur, 2015; Menzies et al, 2017). Enhancing autophagy is a putative therapeutic avenue for neurodegenerative disorders with misfolded protein accumulations (Karabiyik et al, 2017; Menzies et al, 2017; Djajadikerta et al, 2020). ATG9A is the only transmembrane autophagy component and functions at the early steps of autophagosome formation (Young et al, 2006; Webber & Tooze, 2010). In mammalian cells, ATG9A transiently interacts with and contributes to phagophore formation (Orsi et al, 2012; Judith et al, 2019). ATG9A cycles between the Golgi Apparatus, recycling endosomes and plasma membrane, and its trafficking, including endocytosis, is essential for autophagy (Noda et al, 2000; Young et al, 2006; Imai et al, 2010; Puri et al, 2013). Interestingly, genes involved in both autophagy and endocytosis have been associated with neurodegenerative disorders (Alegre-Abarrategui & Wade-Martins, 2009; Schreij et al, 2016; Bandres-Ciga et al, 2019; Vidyadhara et al, 2019; Overhoff et al, 2020). Loss of function of mammalian ATG9A in vivo results in reduced autophagy, highlighting ATG9A's importance in efficient autophagosome formation (Saitoh et al, 2009; Orsi et al, 2012; Popovic & Dikic, 2014; Imai et al, 2016). Brain-specific deletion of *Atg9a* in mice causes accumulation of the autophagy adaptor p62 and of ubiquitinated proteins, and swollen degenerative axons, linking loss of ATG9A to neurodegeneration (Yamaguchi et al, 2018).

[1]Kinases and Brain Development Laboratory, The Francis Crick Institute, London, UK   [2]Mass Spectrometry Proteomics Science Technology Platform, The Francis Crick Institute, London, UK   [3]Randall Centre for Cell and Molecular Biophysics, King's College London, London, UK   [4]Electron Microscopy Science Technology Platform, The Francis Crick Institute, London, UK   [5]Molecular Cell Biology of Autophagy Laboratory, The Francis Crick Institute, London, UK

Correspondence: Sila.ultanir@crick.ac.uk
Flavia Roşianu's present address is MSD R&D Innovation Centre, London, UK

Nuclear Dbf2-related (NDR) kinases NDR1 and NDR2 (NDR1/2) are highly related AGC family serine/threonine kinases that are evolutionarily conserved from yeast to mammals (Hergovich et al, 2006). In various organisms they have been implicated in cellular proliferation (Leger et al, 2018), transcriptional asymmetry (Mazanka et al, 2008), DNA damage response (Qin et al, 2020), innate immunity (Liu et al, 2019), ciliogenesis (Chiba et al, 2013), mitochondrial health (Wu et al, 2013) and neuronal development (Emoto et al 2004, 2006; Ultanir et al, 2012; Rehberg et al, 2014). NDR kinases have also been implicated in autophagy, although it was not clearly established whether they activate or inhibit autophagy as dual deletions of both NDR1 and NDR2 have not been performed in mammals. Furthermore, the mechanisms by which NDR kinases affect autophagy have not been fully elucidated (Amagai et al, 2015; Joffre et al, 2015; Martin et al, 2019). Despite the diverse and essential functions of these highly conserved kinases, their roles in the mammalian brain in vivo have not been explored either, and the mechanisms downstream of NDR kinase activity are poorly understood.

Given the high level of conservation between NDR1 and NDR2, dual deletion of both NDR kinases is essential to study their function and determine their molecular mechanisms. To investigate the roles of NDR1/2 in neurons in vivo, we used *Ndr1* constitutive knockout and *Ndr2*-floxed mice (Schmitz-Rohmer et al, 2015), and knocked out *Ndr2* in excitatory neurons using the NEX-Cre driver (Goebbels et al, 2006). In dual neuronal *Ndr1* and *Ndr2* knockout mice, we observed prominent neurodegeneration in the cortex and hippocampus. Our comparison of hippocampal proteome and phosphoproteome, between control and knockout littermates, identified a major alteration in endocytic pathways and revealed several novel NDR1/2 kinase substrates that contain the previously reported HXRXXS* motif. We validated Ras association (RalGDS/AF-6) and pleckstrin homology domains 1 (Raph1), a.k.a. lamellipodin (Lpd), as a novel NDR1/2 substrate. We show that both NDR1/2 and Raph1 regulate neuronal endocytosis. In the absence of NDR1/2, p62 and ubiquitin accumulate in mouse neurons. Furthermore, autophagosome numbers are reduced in NDR1/2 knockout neurons, and autophagic clearance is affected in primary neurons depleted of NDR1/2 kinases, overall showing that NDR1/2 are required for the maintenance of neuronal protein homeostasis. Finally, deletion of NDR1/2 in adult mice also results in a similar phenotype, indicating that NDR1/2 are needed for protein homeostasis not only during neuronal development. Mechanistically, we show that NDR kinases are critical regulators of clathrin-mediated endocytosis (CME) and ATG9A trafficking, revealing a novel kinase regulatory pathway that is essential for autophagy in neurons. We conclude that NDR1/2 are required for ATG9A trafficking, autophagy and for the maintenance of neuronal homeostasis to prevent neurodegeneration.

## Results

### Dual deletion of NDR1/2 kinases in excitatory neurons causes neurodegeneration and reduces mouse survival

To investigate the roles of NDR1/2 kinases in mammalian neurons in vivo, we generated a mouse model in which both *Ndr1* (also known as *stk38*) and *Ndr2* (also known as *stk38l*) were deleted in excitatory post-mitotic neurons from the cortex and hippocampus.

Individual *Ndr1* or *Ndr2* full knockout mice are viable and fertile, but dual *Ndr1/Ndr2* knockout mice are embryonically lethal, indicating that the two isoforms compensate for each other's function (Schmitz-Rohmer et al, 2015), likely owing to their 87% amino acid identity (Tamaskovic et al, 2003). Constitutive *Ndr1* knockout (*Ndr1*^KO) and floxed *Ndr2* (*Ndr2*^flox) mice (Schmitz-Rohmer et al, 2015) were crossed with mice expressing the Cre recombinase under the control of the NEX driver (Goebbels et al, 2006), which is specific for pyramidal neurons of the cortex and hippocampus. Our experimental crosses gave litters with four possible genotypes: *Ndr1*^KO/+ *Ndr2*^flox/+ NEX^Cre/+ (control), *Ndr1*^KO/KO *Ndr2*^flox/+ NEX^Cre/+ (NDR1 KO), *Ndr1*^KO/+ *Ndr2*^flox/flox NEX^Cre/+ (NDR2 KO) and *Ndr1*^KO/KO *Ndr2*^flox/flox NEX^Cre/+ (NDR1/2 KO). As previously reported (Cornils et al, 2010), NDR1 KO mice were viable and fertile, and exhibited normal brain development, and the same was true of the NDR2 KO mice, which lacked NDR2 in excitatory neurons (Fig S1A and B). NDR1/2 KO mice were also viable, but had significantly lower weights and reduced survival rate compared with littermates (Fig 1A and B). Weight and survival rate were not affected in NDR1 or NDR2 individual KO mice (Fig 1A and B), further confirming their mutual compensation.

In NDR1/2 KO mice, cortical thickness was unchanged at post-natal day 20 (P20) (Fig 1C), but at 12 wk of age, the cortex thickness was significantly reduced (Fig 1D). Immunostainings of the neuronal nuclear marker CTIP2 at 12 wk showed that CTIP2-positive neurons in deep cortical layers were largely intact, while upper layers (above CTIP2-positive layers) exhibited reduced thickness, indicating loss of neurons in these areas (Fig 1D). Individual NDR1 or NDR2 KOs were not affected, even as late as 20 wk of age (Fig S1A). Astrocyte and microglial activation are hallmarks of neurodegeneration in humans and mouse models. The astrocytic marker GFAP was highly increased in NDR1/2 KO mice (Figs 1E and F and S1C), and additionally, microglia exhibited a hypertrophied morphology associated with "reactive" cells, in contrast to the ramified "resting" microglial cells (Ransohoff, 2016) present in control mice (Fig 1G). Individual NDR1 KO and NDR2 KO mice did not have increased GFAP levels, despite a clear reduction in the protein levels of NDR1 and NDR2, respectively (Figs 1F and S1B and C). H&E stainings from the upper layers of the cortex of NDR1/2 KOs revealed neurons with a deeply eosinophilic cytoplasm and small condensed nuclei, indicative of apoptosis or necrosis (Fig S1D). To a lesser extent, necrotic neurons were also observed in the hippocampus (Fig S1D).

To explore the morphology of these degenerating neurons, we crossed NDR1/2 KO mice with Thy1-YFP mice, which express YFP sparsely in pyramidal neurons (Feng et al, 2000). In the hippocampus of 12-wk-old mice, the CA1 cell body layer was disorganised, whereas hippocampal thickness (Fig 1H) and apical dendrite length (Fig S1E) were significantly reduced, indicating neuropil loss. This decrease in apical dendrite length happened gradually, as mice aged, since at P20 no such difference was observed between controls and NDR1/2 knockouts (Fig S1F). Overall, our data show that dual, but not individual deletion of NDR1 and NDR2 in neurons leads to neurodegeneration of upper cortical layers and to a lesser extent the hippocampus.

NDR1/2 play a role in dendrite and spine morphogenesis and actin regulation (Geng et al, 2000; Emoto et al, 2004; Ultanir et al, 2012). NDR1/2 knockout neurons have small protrusions and

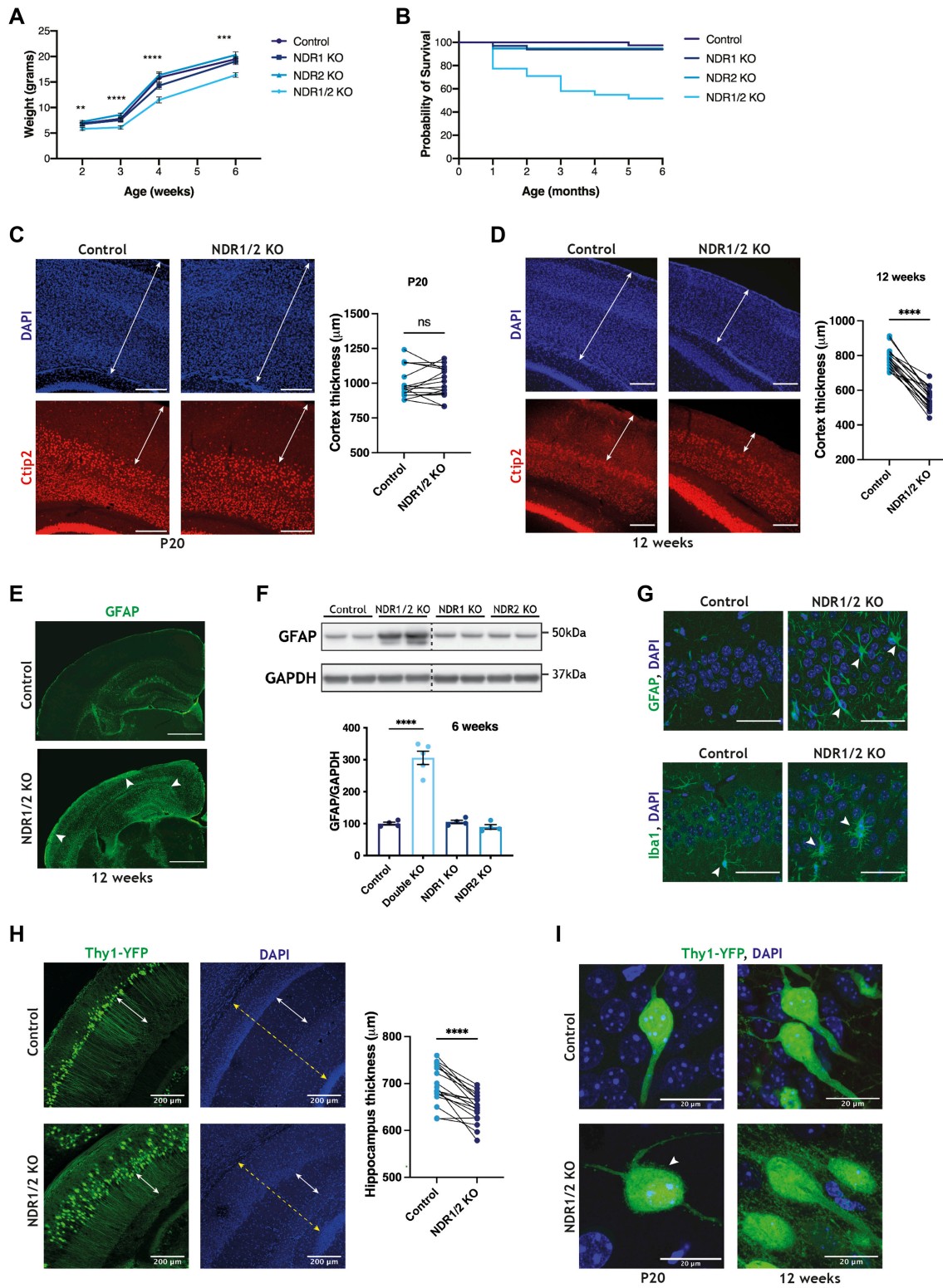

**Figure 1. Dual loss of NDR kinases in neurons leads to neurodegeneration.**
**(A)** Line graph showing the average weight of NDR1 KO, NDR2 KO, NDR1/2 KO and control mice up to 6 wk of age. The differences in weight between the genotype groups at each time-point were analysed using ordinary one-way ANOVAs or Kruskal–Wallis tests. n = 20–30 mice/group. **(B)** Graph illustrating the probability of survival of NDR1 KO, NDR2 KO, NDR1/2 KO and control mice up to 6 mo of age. n = 20–30 mice/group. **(C, D)** Immunofluorescence staining of Ctip2 in brain slices of NDR1/2 KO and control mice at P20 or 12 wk of age. White arrowed lines in the DAPI images show the thickness of the cortex, and white arrowed lines in the Ctip2 images show the thickness of the upper layers I–IV of the cortex. Scale bars: 200 μm. The graphs show quantifications of cortex thickness, and the data were analysed using paired t tests. n = 18

membrane ruffles on the cell body and dendrites, in contrast to the smooth plasma membrane of control cells at postnatal day 20 (Fig 1I). By 12 wk, these protrusions become more exuberant and occupy the area in between the cells, as the knockout neurons lose the tightly packed distribution of cell bodies, characteristic of control neurons (Fig 1H and I). However, we observed no differences in the total levels of key synaptic proteins in the cortex of NDR1/2 KO mice (Fig S1G), although Thy1-expressing layer 5 neurons exhibited membrane protrusions similar to CA1 neurons (Fig S1H). Transmission electron microscopy (TEM) assessment of brain slices from NDR1/2 KO mice confirmed the changes in cellular morphology seen in Thy1-YFP mice and revealed changes in mitochondria, which were rounder and more fragmented, compared to control brains (Fig S1I). Such changes in mitochondrial morphology have been previously reported in apoptotic cells and neurodegenerative conditions (Knott et al, 2008; Su et al, 2010), and could represent an early step during the apoptosis process, but could also be a result of the more direct role of NDR kinases in mitochondrial quality control (Wu et al, 2013). In this study we aimed to uncover the mechanisms through which loss of both NDR1/2 kinases impinges on neuronal health in mice.

### Proteomics analyses of NDR1/2 KO hippocampi show changes in endocytosis-related proteins and reveal novel substrates

To gain mechanistic insight into the function of NDR1/2, we employed a mass spectrometry approach to identify the substrates of NDR that could mediate their roles, and to assess changes in the levels of proteins in NDR1/2 KO brains at a global scale. Because of the high GFAP expression in the cortex of NDR1/2 KO mice as early as P20 (Fig S1C), we used hippocampus samples for the proteomics analysis. Astrocytes are not targeted by the Cre recombinase and could confound our results, since we were interested in neuron-specific changes that happen directly downstream of NDR1/2 loss. Briefly, hippocampi were dissected from NDR1/2 KO and control littermates. A total of 5 control and 5 NDR1/2 KO samples were trypsin-digested and labelled with a tandem mass tag (TMT) 10-plex reagent (Jiang et al, 2017). For phosphoproteome analysis, phosphopeptides were enriched using the titanium dioxide and Fe-NTA methods (Jiang et al, 2017) (Fig 2A).

The total proteome analysis identified 7456 proteins, out of which 408 were significantly up-regulated and 255 were down-regulated in the NDR1/2 KO brains (Fig 2B and Table S1). Remarkably, the top two most reduced proteins were NDR1 and NDR2, as expected (Fig 2B). Interestingly, Mob2, a conserved interactor and activator of NDR kinases (Weiss et al, 2002; Bichsel et al, 2004), was also markedly down-regulated in the absence of its interacting

partners NDR1/2. The top up-regulated proteins include several enzymes involved in lipid metabolism, such as Lpcat, Acaca and Fasn, and the cell adhesion molecules L1cam and Chl1 (Fig 2B). Accordingly, when running gene enrichment analyses using the online tool DAVID (Huang et al, 2009a, 2009b), "cell adhesion" comes up as a biological process highly enriched in the list of up-regulated proteins. In addition, "membrane"-related terms are also enriched in the cellular compartment category, likely matching the changes in lipid metabolism (Fig 2C and Table S2). Another biological process associated with the list of up-regulated proteins that stands out is "autophagy" (Fig 2C and Table S2). The significantly down-regulated proteins were associated with biological processes and cellular compartments linked to myelination (Fig S2A and Table S3). Interestingly, a reduction in myelination can be indicative of axonal degeneration, which has been previously reported in mouse models with neuron-specific deletions of autophagy-related proteins (Komatsu et al, 2007b; Liang et al, 2010; Zhao et al, 2013). In terms of KEGG pathways and human phenotype ontology, the enriched terms within these categories describe features or diseases reminiscent of the neurodegeneration phenotype observed in NDR1/2 KO brains, such as "neurofibrillary tangles," "dementia" and "cerebral inclusion bodies" (Fig S2B and Table S4).

To identify signalling differences between control and NDR1/2 KO mice, we compared their hippocampal phosphoproteomes. TMT labelling mass spectrometry identified 39620 unique phosphorylation sites, with numerous significant differences between the two genotypes (Fig 2D). We used the established NDR kinase consensus motif HXRXXS/T*, where * denotes phosphorylation (Mazanka et al, 2008; Ultanir et al, 2012), to filter the data for potential NDR substrates. Additonally, a stringent significance cut-off of adjusted $P < 0.005$ was applied to find the most robustly reduced phosphosites in the absence of NDR1/2 kinases (Fig 2D and Table S5). This produced a list of 11 putative NDR1/2 substrates (Fig 2D and E). Three of them were previously reported to be direct substrates of NDR1/2 in mouse brains, namely PI4KB, Rab11fip5 and AAK1 (Ultanir et al, 2012), providing validation to our experiment. These substrates play roles predominantly in endomembrane trafficking (Fig 2E). To assess the implications of all changed phosphorylations, we performed gene enrichment analyses using the genes corresponding to altered phosphorylation sites with an adjusted $P < 0.005$ (Fig 2F). Interestingly, "endocytosis" was the most significant biological process enriched in this list of genes, providing a link between NDR1/2 kinases and this cellular process (Figs 2F and S2C and Table S6). Furthermore, the putative direct substrates Raph1, AAK1 and SYNG all regulate clathrin-mediated endocytosis (CME) (Conner & Schmid, 2002; Hirst et al, 2005; Vehlow et al, 2013), and multiple proteins with roles in endocytosis were also significantly

---

measurements from three mice/genotype. **(E)** Immunofluorescence staining of GFAP in brain slices of 12-wk-old NDR1/2 KO and control mice. White arrows show areas with increased GFAP signal in NDR1/2 KO mice. **(F)** Western blot analyses of GFAP levels in lysates from the cortex of 6-wk-old mice. GAPDH was used as a loading control. The graphs show quantifications of the GFAP bands normalised against the GAPDH levels, and the data were analysed using an ordinary one-way ANOVA with Tukey's post hoc test. n = 3–5 mice/group. **(G)** Immunofluorescence staining of GFAP and the microglial marker Iba1 in the CA1 area of the hippocampus. White arrows indicate cells expressing the above-mentioned markers. Scale bars: 50 μm. **(H)** Images from brain slices of 12-wk-old Thy1-YFP–expressing mice in the CA1 area of the hippocampus. White arrowed lines indicate the stratum radiatum, where CA1 neuron dendrites are visible in YFP. The graph shows quantification of the hippocampal thickness (marked by the dashed yellow line—including stratum oriens, the CA1 cell body area, stratum radiatum and stratum lacunosum-moleculare), and the data were analysed using a paired *t* test. n = 18 measurements from three mice/genotype. **(I)** Images from brain slices of Thy1-YFP–expressing mice in the CA1 cell body layer. The white arrow shows membrane protrusions present in NDR1/2 knockout neurons.
Source data are available online for this figure.

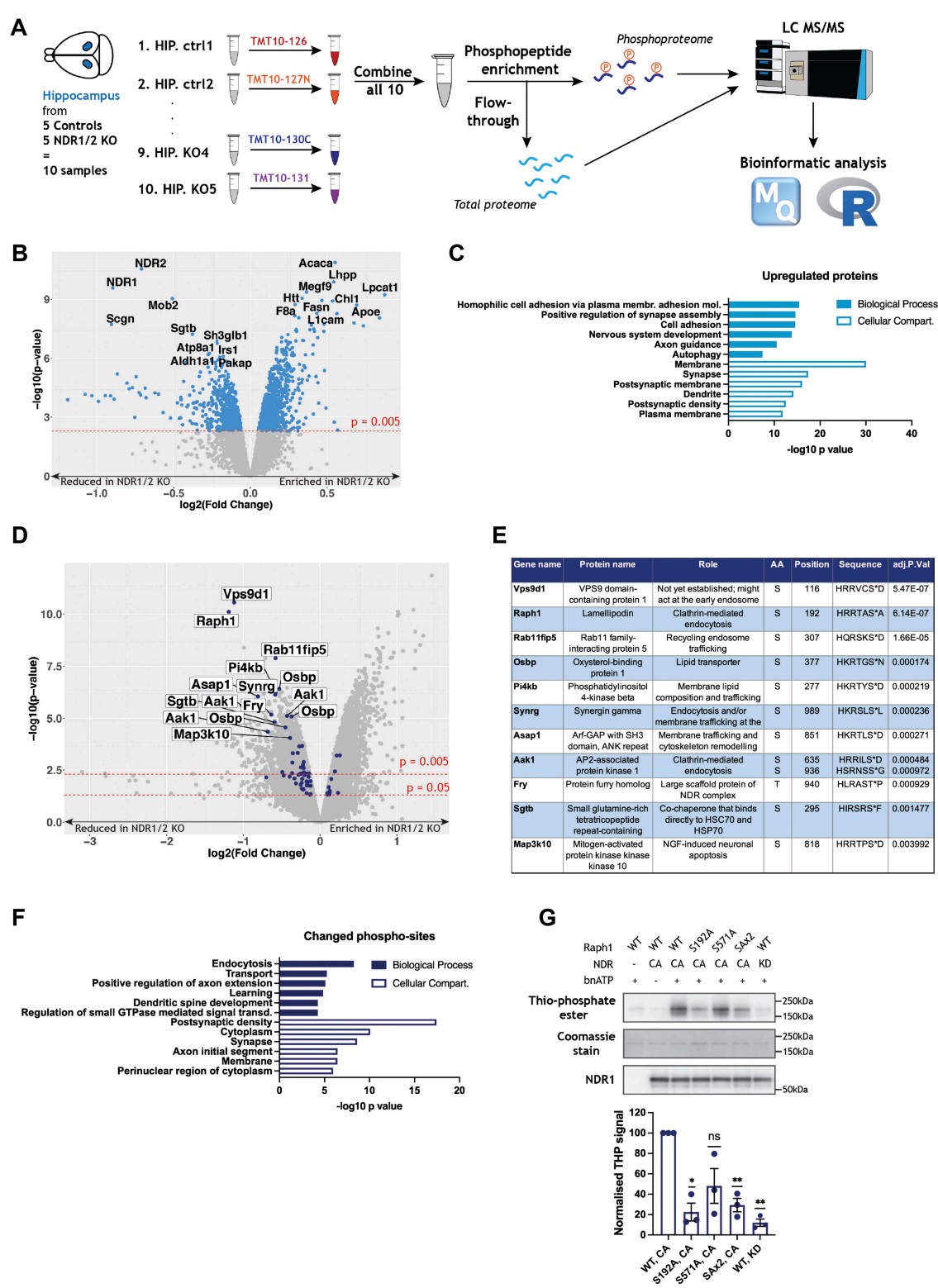

**Figure 2. Quantitative proteomics and phosphoproteomics indicate altered endocytosis and identify Raph1/Lpd as an NDR kinase substrate.**
**(A)** Workflow of tandem mass tag labelling of hippocampus samples from the brains of NDR1/2 knockout and control mice with subsequent phospho-enrichment and mass spectrometry analysis of the total and the phosphoproteome. MaxQuant software and R coding were used for the bioinformatic analysis. **(B)** Volcano plot of the difference in protein levels between control and NDR1/2 knockout mice. Each point represents one protein. The x-axis shows $\log_2$-transformed fold change and the y-axis shows significance by $-\log_{10}$-transformed $P$-value, obtained by linear models for microarray data. All significantly changed proteins are highlighted in blue. A protein is considered significantly changed if its $P$-value is < 0.005, with the red line marking the difference between significantly and non-significantly changed proteins. The top 10 up-regulated or down-regulated proteins in the NDR1/2 knockout brains compared to the control brains are labelled in black. **(C, F)** Results from gene enrichment analyses

up- or down-regulated in the total proteome of NDR1/2 KO mice (Fig S2D). We also assessed the localisation and distribution of endophilin A and clathrin heavy chain, which were not present/altered in our mass spectrometry dataset, but are key players in endocytosis. Endophilin A is also a binding partner of the NDR substrate candidate Raph1/Lpd (Vehlow et al, 2013; Boucrot et al, 2015). Immunofluorescence staining of these markers in mouse brain slices however, found no discernable differences between controls and NDR1/2 knockouts (Fig S2E). Collectively, proteomics analyses strongly indicate that NDR1/2 kinases play a role in endomembrane trafficking and endocytosis.

We decided to validate one of the most promising substrate candidates, Raph1/Lpd, due to its roles in CME and fast endophilin-mediated endocytosis (Vehlow et al, 2013; Boucrot et al, 2015; Chan Wah Hak et al, 2018). Similarly to NDR kinases, Lpd is also required for neuronal dendritic arborisation (Tasaka et al, 2012). Lpd harbours a Ras association (RA) domain followed by a pleckstrin homology (PH) domain (Fig S2F) (Krause et al, 2004). The phosphosite identified with mass spectrometry, Ser192, precedes the RA domain, and we noted that another residue with the NDR consensus motif, Ser571, is located closely after the PH domain (Fig S2F). We purified wild-type, single phosphomutant and double phosphomutant Lpd (Fig S2G) and performed in vitro kinase assays using constitutively active (CA) NDR1 or kinase-dead (KD) NDR1 (Figs 2G and S2H). We found that NDR1 specifically phosphorylates Raph1/Lpd on S192, whereas S571 is not consistently phosphorylated in vitro (Fig 2G). Phosphorylation of S192 was confirmed in HEK293T cells with a phospho-specific antibody raised against a phosphopeptide containing this site (Fig S2I). Overall, our phosphoproteomics results indicate that NDR1/2 play a role in endocytosis and identify Raph1/Lpd as a novel substrate.

### NDR1/2 depletion reduces endocytosis, partially through Raph1/Lpd

Because of the significant changes in endocytosis-related phosphorylation events in our proteomics analysis, we decided to assess whether NDR1/2 depletion impacts endocytosis in cultured primary neurons. Transferrin receptor (TfR) is a transmembrane protein that internalises transferrin (Tf)-bound iron and subsequently gets recycled through the endosomal pathway back to the plasma membrane to ensure further iron uptake (Aisen, 2004). Tf-based assays are used to study CME. We performed labelled Tf-uptake experiments, using Tf conjugated to Alexa Fluor 568 (Tf-568), in hippocampal neurons previously infected with either scramble

shRNA or dual NDR1 shRNA and NDR2 shRNA, which robustly knocked down their target genes (Fig S3A). After 20 min of incubation with Tf-568 (pulse), the Tf-containing media was replaced with maintenance media devoid of Tf-568, for 20 min or 60 min (chase), to allow for recycling of Tf. Tf-568 signal was significantly reduced in NDR1/2 shRNA neurons after pulse, at time-point 0 min, indicating that endocytosis is less efficient in the absence of NDR1/2 kinases (Fig 3A, image panel and scatterplot). After the 20- and 60-min Tf chase, scramble shRNA–expressing neurons recycled Tf more efficiently than NDR1/2 shRNA–expressing neurons, indicating that NDR1/2 depletion also impairs recycling efficiency (Fig 3A, line graphs).

To confirm an impairment in TfR trafficking, we assessed the levels of TfR at the plasma membrane, using surface biotinylation experiments. Biotin labelling of surface proteins was carried out on ice to block endocytosis. Biotinylated proteins were subsequently isolated and their levels measured using Western blotting. The surface levels of TfR were higher in neurons in which NDR1/2 had been knocked down, when normalised to total TfR levels (Fig 3B). Surprisingly, the surface levels of the adhesion molecule Chl1, a regulator of integrin signalling (Buhusi et al, 2003) that was significantly up-regulated in the brains of NDR1/2 KO mice (Fig 2B), were reduced in NDR1/2-depleted neurons (Fig 3B). The NDR kinase homolog trc is required for adhesion of dendrites to epithelia via integrin signalling in flies (Han et al, 2012), so it is possible that mammalian NDR1/2 has a similar function, mediating neuronal morphology and cell body layering. By contrast, surface levels of the AMPA-type glutamate receptor subunit GluR1 were not altered after NDR1/2 depletion (Fig 3B), in agreement with no apparent differences in the levels of synaptic markers in NDR1/2 KO brains (Fig S1F). These results suggest that both TfR and Chl1 trafficking, and consequently their surface levels, are impaired in the absence of NDR1/2.

To check if TfR trafficking is altered in vivo in NDR1/2 KO mice, we assessed the localisation of endogenous TfR in brain silces, using immunohistochemistry. Compared to the diffuse and less discrete TfR signal in control mice, NDR1/2 KO brains contained distinct TfR puncta, which accumulated with age, indicative of a blockage in TfR recycling (Fig 3C). At P20, few TfR puncta were present in NDR1/2 KOs, but these increased to a striking number by 12 wk of age (Fig 3C, scatter plot), revealing that TfR accumulation is an early impairment that increases progressively. The upregulation in TfR was confirmed with Western blots at 6 wk of age (Fig S3B). To further characterise TfR puncta in NDR1/2 KOs, we co-stained TfR with endosomal markers. The retromer component VPS35 (Seaman,

run using the online tool Database for Annotation, Visualization and Integrated Discovery (DAVID), with the list of genes corresponding to all significantly up-regulated proteins (C) or all significantly changed phosphopeptides (F). The x-axis shows the P-value or EASE score generated by DAVID to show how enriched a term associated with a list of genes is. The top 6 most enriched terms from each category are represented. **(D)** Volcano plot showing the difference in phosphopeptide levels between control and NDR1/2 knockout mice. Each point represents one phosphopeptide. The x-axis shows $\log_2$-transformed fold change and the y-axis shows significance by $-\log_{10}$-transformed P-value, obtained by linear models for microarray data. All phosphopeptides above the higher red line have a $P < 0.005$ and all phosphopeptides above the lower red line have a $P < 0.05$. The blue dots represent phosphopeptides with the already established NDR consensus motif HXRXXS/T. All phosphopeptides with the NDR consensus motif and an adjusted $P <0.005$ have been labelled. **(E)** Table with the roles, specific phosphorylation site and phosphorylated sequence of all NDR substrate candidates with an adjusted $P < 0.005$. **(G)** Representative Western blots showing thiophosphate ester and NDR1 levels in an in vitro kinase assay. Total Raph1 levels are shown by Coomassie staining. The bar graph shows quantification of Raph1 thiophosphorylation, normalised to total Raph1 levels and expressed as a percentage of the Raph1-WT/NDR-CA thiophosphate ester level. The data were analysed using a one-sample $t$ test, n = 3 independent experiments.
Source data are available online for this figure.

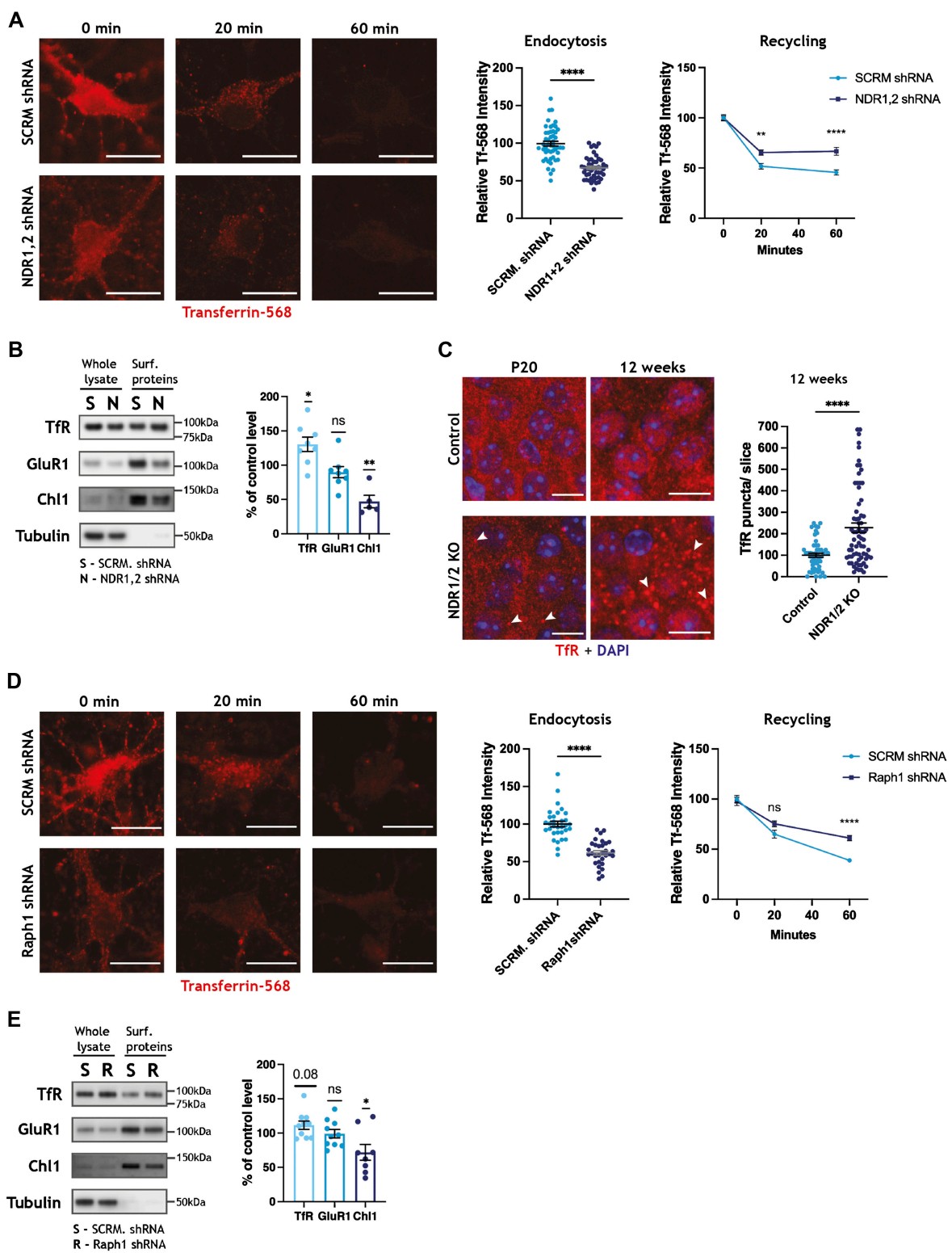

**Figure 3. NDR kinases and their substrate Raph1/Lpd are required for endocytosis.**
**(A, D)** Representative images of DIV11 rat hippocampal primary neurons incubated with transferrin–Alexa Fluor 568 (Tf-568) for 20 min (pulse) and chased for 0, 20, or 60 min in complete media. Scale bars: 10 μm. **(A, D)** Neurons were infected with scramble (SCRM) shRNA lentivirus or lentiviral vectors expressing NDR1 and NDR2 shRNAs (NDR1,2 shRNA) (A) or Raph1/Lpd shRNA (D) and transfected with an empty EGFP-expressing plasmid (not depicted). The scatterplots represent the amount of Tf-568 present in neurons at the end of pulse (time-point: 0 min), normalised to the cell area (using the EGFP cell fill), and the data were analysed with unpaired *t* tests. The line graphs represent the quantification of the remaining cellular Tf-568 after the chase period, when the total amount of transferrin endocytosed was normalised to 100. The

2012), which plays a role in tubulation of endosomes, also displayed a similar discrete accumulation and significantly co-localised with TfR puncta in NDR1/2 KOs (Fig S3C). Taken together, these results show that TfR and VPS35-positive endosomal compartments accumulate in NDR1/2 KO neurons, highlighting defects in endosomal trafficking.

Our phosphoproteome assessment identified several NDR1/2 target substrates, which could contribute to the membrane trafficking defects highlighted above. We selected Raph1, a known mediator of endocytosis (Vehlow et al, 2013), and tested if it plays a role in TfR trafficking, using lentivirus to knock down Raph1 in primary neurons (Fig S3D). Depletion of neuronal Raph1 resulted in less Tf-568 being endocytosed after the 20 min pulse (Fig 3D, image panel and scatterplot) and less Tf-568 being recyled after the 60 min chase (Fig 3E, line graphs), mimicking loss of NDR1/2. Interestingly, the surface levels of TfR also showed a slight increase in Raph1 shRNA neurons, but this change was not significant (Fig 3E). Chl1 surface levels were significantly reduced, while surface GluR1 was not affected in the absence of Raph1 (Fig 3E), replicating the effect of NDR1/2 knockdown. To test if NDR1/2 and Raph1 can interfere with each other's function, we transfected Raph1 in NDR1/2 shRNA–expressing primary neurons. We find that Raph1 can partially ameliorate the Tf-568 endocytosis deficit caused by loss of NDR1/2, indicating that Raph1 is functioning downstream of NDR1/2 in promoting endocytosis (Fig S3F). Overall, these results suggest that NDR1/2 and their substrate Raph1 regulate endocytosis of TfR and membrane trafficking.

To further investigate the role of NDR1/2 phosphorylation on the S192 site of Raph1, we transfected rat primary hippocampal neurons with 3×FLAG-tagged Raph1 and stained for the FLAG tag. We found that both WT-Raph1-3×FLAG and phosphomutant S192A-Raph1-3×FLAG localise to the cellular membrane, with no clear changes in distribution between the two constructs. The FLAG signal was confirmed to be specific for Raph1 with a Raph1 antibody (Fig S3E), indicating that phosphorylation is unlikely to alter Raph1 localisation.

## Autophagy is impaired in neurons that lack NDR1/2 kinases

Our NDR1/2 KO mouse model presented striking similarities to brain-specific autophagy-defective mouse models, such as the timing of neuronal loss and specific loss of upper cortical layers (Hara et al, 2006; Komatsu et al, 2006). Furthermore, several autophagy-related proteins are up-regulated in our proteomics dataset and changes in endocytosis can impact autophagosome formation (Puri et al, 2013; Popovic & Dikic, 2014; Tooze et al, 2014). Given the reported role of NDR1/2 kinases in autophagy, we decided to test if autophagy is impaired in our mouse model. The autophagy

adaptor p62 binds ubiquitinated proteins and targets them to autophagosomes, while also being itself degraded by autophagy (Komatsu et al, 2007a; Pankiv et al, 2007). Immunostainings in brain sections revealed a prominent accumulation of p62 in NDR1/2 KOs by 12 wk of age (Fig 4A). On Western blots, it became evident that at P20 the increase in p62 is not yet significant, but by 6 wk of age significantly more p62 is present in NDR1/2 KO brains (Fig 4B), indicative of a gradual age-dependent upregulation in this marker. Concurrently, the autophagy adaptor NBR1 was also increased in NDR1/2 KO mouse brains on Western blots (Fig S4A), and the levels of the large scaffold protein Alfy (WDFY3), another autophagy adaptor involved in degradation of ubiquitinated proteins, were increased in our proteomics dataset (Table S1). These results show that multiple autophagy adaptors are upregulated in NDR1/2 KO mouse brains, pointing towards a more generalised impairment of the pathway. Considering the significant astrogliosis in NDR1/2 KO brains, we sought to establish if the observed accumulation in p62 was specifically present in neurons or in non-neuronal cells. To this end we used Thy1-YFP–expressing mice and quantified p62 puncta in YFP-positive cell bodies. The results revealed a significant increase in p62 puncta in the soma of NDR1/2 KO neurons at 12 wk of age (Fig 4C). Since our stainings show a widespread p62 accumulation at the level of the CA1 area (Fig 4A), this effect is not specific to Thy1-YFP neurons, but to all CA1 neurons. We also assessed the presence of p62 in glial cells with co-stainings, and found that the increase in p62 was specific to neurons and not present in GFAP-positive astrocytes (Fig S4B) or Iba1-positive microglia (Fig S4C). Furthermore, stainings from brain sections also showed a marked increase in cytoplasmic ubiquitin in NDR1/2 KO neurons, and ubiquitin puncta co-localised with p62, revealing an accumulation in ubiquitinated proteins targeted to the autophagy pathway (Fig 4D). The increase in ubiquitinated proteins was also confirmed with Western blots in the brains of 6-wk-old NDR1/2 KO mice (Fig S4D). These results agree with high levels of ubiquitin and p62 in autophagy-impaired mouse models (Hara et al, 2006; Komatsu et al, 2006; Kuijpers et al, 2021).

The accumulation in p62 and ubiquitinated proteins could result from a defect in autophagosome formation and subsequent impairment in the clearance of these proteins. Consequently, we assessed whether the levels of the autophagosome marker LC3 were changed in NDR1/2 KO neurons. We detected autophagosomes in YFP-positive neuronal cell bodies, and found that the number of LC3 puncta was significantly lower in NDR1/2 KO neurons when compared to controls (Fig 4E). These results indicate that autophagosome formation is reduced in the absence of NDR kinases.

If autophagosomes still form, but ubiquitinated proteins are not cleared adequately, we wondered if this is a result of inefficient targeting of ubiquitinated proteins to the autophagy pathway or a

data were analysed using a mixed-effects analysis with Šidák's multiple comparisons test, n > 30 cells/condition from three independent experiments. **(B, E)** Western blot analyses of surface biotinylation experiments in DIV12 rat cortical neurons, infected with lentiviral vectors expressing scramble (SCRM) shRNA and NDR1,2 shRNA (B) or SCRM shRNA and Raph1 shRNA (E). Surface protein levels were normalised against input. Bar graphs show surface protein levels in knockdown conditions, expressed as a percentage of the corresponding SCRM shRNA control level. The data were analysed using a one-sample $t$ test. n = 6 samples/condition from three independent experiments. **(C)** Immunofluorescence staining of transferrin receptor (TfR) in the CA1 area of the hippocampus in brain slices from P20 and 12-wk-old NDR1/2 knockout and control mice. White arrows indicate TfR-positive puncta. Scale bars: 10 μm. The scatterplot shows the number of TfR puncta/slice in non-nuclear areas. The data were analysed using a Mann–Whitney test. n = 52 control and 72 NDR1/2 KO slices from three mice per genotype.
Source data are available online for this figure.

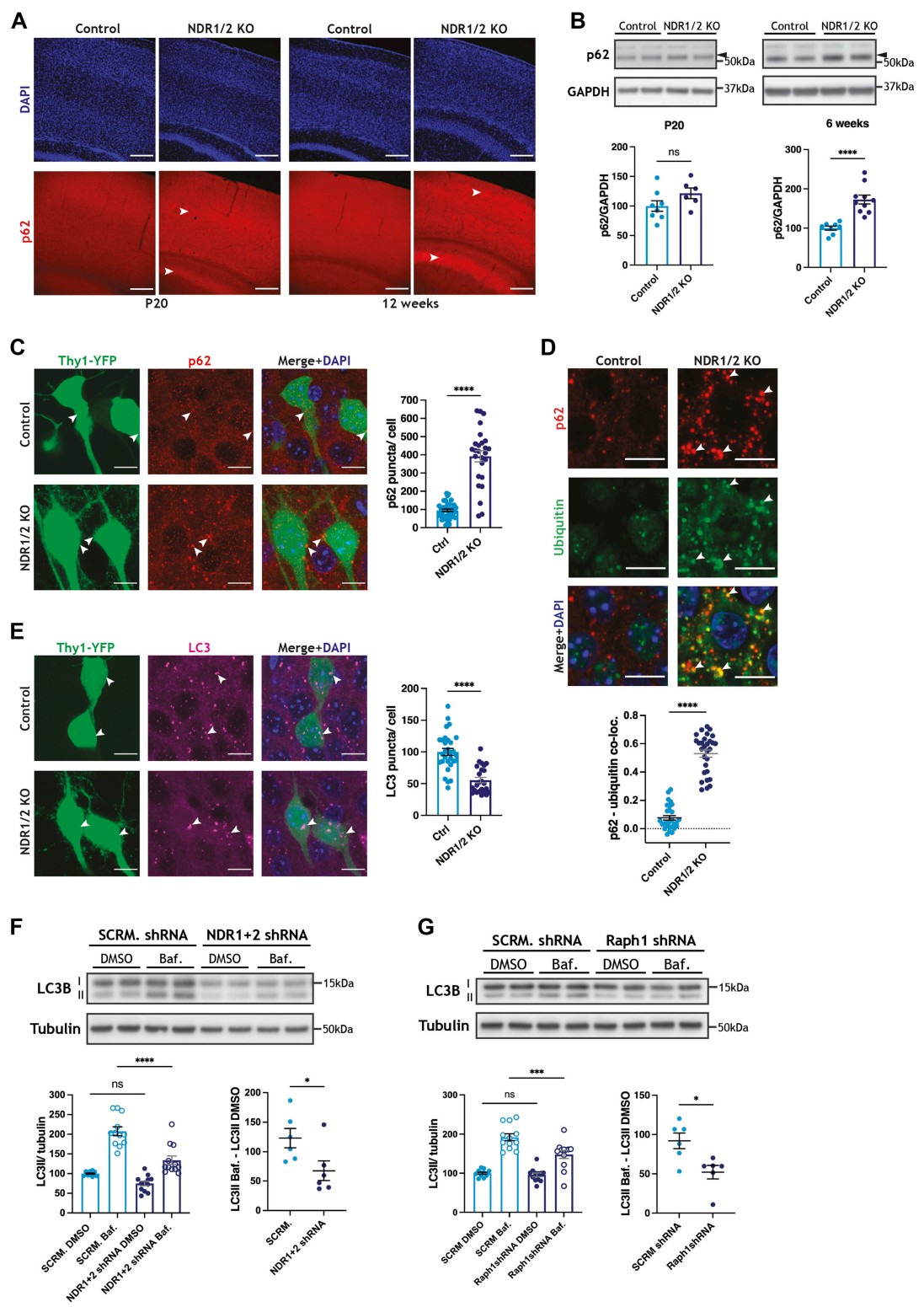

**Figure 4. Autophagy is impaired in NDR1/2 KO mice.**
**(A)** Immunofluorescence staining of the autophagy receptor p62 in brain slices of NDR1/2 KO and control mice at P20 and 12 wk of age. White arrows indicate areas of increased p62 signal. Scale bars: 200 μm. **(B)** Western blot analyses of p62 levels in lysates from the cortex of P20 and 6-wk-old NDR1/2 KO and control mice. GAPDH is used as a loading control. The bar graphs show quantifications of the p62 bands normalised against the GAPDH levels, and the data were analysed using unpaired *t* tests. n = 3–5 mice/group, two technical replicates. **(C, E)** Immunofluorescence staining of p62 (C) and LC3 (E) in the CA1 area of the hippocampus in brain slices from 12-wk-old Thy1-YFP–expressing mice. Scale bars: 50 μm. **(C, E)** Bar graphs show the number of p62 (C) or LC3 (E) puncta in neurons, normalised against the cell body area corresponding to the YFP signal. The data were analysed using a Mann–Whitney test for p62 and an unpaired *t* test for LC3. n = 38 control and 27 NDR1/2 KO neurons from two mice per

failure of autophagosomes to fuse with lysosomes. We performed co-stainings of p62, LC3 and the lysosomal marker LAMP2 in brain sections. In both controls and NDR1/2 KOs, almost all LC3 puncta co-localised with p62 and LAMP2 (Fig S4E), indicating that LC3 staining represents autolysosomes in the soma. Larger cytoplasmic p62 puncta co-localised with LC3 and LAMP2 in controls, showing targeting of p62 to these compartments (Fig S4E). Some p62 also co-localised with LC3 and LAMP2 in NDR1/2 KO neurons, (Fig S4E, white arrows), showing targeting of p62 to autolysosomes, but there were a substantial amount of p62 puncta that did not overlap with either LC3 or lysosomes (Fig S4E, yellow dotted circles). Considering the gradual accumulation of p62 we observed at P20, 6 wk and 12 wk of age, we suggest that autophagy functions at reduced levels, which results in accumulation of ubiquitinated proteins in NDR1/2 KO neurons over a longer period.

To test this hypothesis, we treated primary neurons with Bafilomycin A1, a drug that inhibits autophagosome–lysosome fusion, blocking autophagosome clearance and enabling assessment of formation rates (Mizushima & Yoshimori, 2007; Rubinsztein et al, 2009). We assessed autophagosome formation in rat primary cortical neurons infected with lentiviral vectors expressing NDR1 shRNA and NDR2 shRNA or a scramble shRNA control. Bafilomycin A1 treatment resulted in a significant increase in lipidated LC3 (LC3II) levels in both control neurons and NDR1/2-depleted neurons (Fig 4F), indicating that autophagosomes formed in both conditions. However, LC3II levels after Bafilomycin A1 treatment were significantly lower in the NDR1/2 shRNA–infected neurons, and the amount of LC3II that accumulated within the treatment time (LC3II Baf.–LC3II DMSO) was also reduced (Fig 4F). These results show that both the formation and degradation of autophagosomes are reduced in NDR1/2-depleted neurons. LC3I levels were also reduced in NDR1/2 KO neurons (Fig S4F), and this may be due to indirect mechanisms downstream of NDR1/2, which impinge on LC3I levels. We next assessed if the role of NDR was dependent on its kinase activity. For this, we transfected HEK293T cells with either constitutively active NDR1 (NDR1-CA) or a kinase-dead version of NDR1 (NDR1-KD) (Ultanir et al, 2012), and treated them with Bafilomycin A1. Although basal levels of LC3II were not changed in the NDR1-CA– or NDR1-KD–expressing cells, the levels of LC3II post-Bafilomycin treatment were significantly higher with NDR1-CA and reduced with NDR1-KD (Fig S4G). These results suggest that the kinase activity of NDR1 is necessary for efficient autophagy.

Consequently, we tested if knockdown of the newly validated NDR substrate Raph1 could mimic some of the effects of NDR1/2 depletion on autophagosome levels. Knockdown of Raph1 in neurons also resulted in reduced LC3II accumulation upon Bafilomycin A1 treatment (Fig 4G), without any changes in LC3I (Fig S4H),

indicating that Raph1 could act downstream of NDR1/2 to mediate its role in autophagy. These results show that NDR1 and NDR2 are required for efficient autophagy in neurons, and via their kinase activity, NDR kinases are sufficient to enhance autophagy in mammalian cells.

## ATG9A trafficking and localisation are altered in NDR1/2 KO mice

TfR-positive recycling endosomes have been linked to the autophagy pathway (Lamb et al, 2013; Puri et al, 2013; Tooze et al, 2014; Soreng et al, 2018). Early-acting key autophagy proteins such as ATG9A and ULK1 can be found on TfR-positive recycling endosomes, and Tf-containing recycling endosomal membranes can contribute to the formation of new autophagosomes (Longatti et al, 2012). In mammalian cell lines, ATG9A can be present at the plasma membrane (Puri et al, 2013; Zhou et al, 2017), where it gets internalised via CME and delivered to the early endosome. From here, it is sorted to the recycling endosome, before reaching autophagosome initiation sites, to contribute membrane to autophagosome precursors (Puri et al, 2013). A block in endocytosis impairs autophagy and results in ATG9A redistribution from perinuclear areas to areas close to the plasma membrane, concomitant with a reduction in ATG9A co-localisation with either Golgi or recycling endosome markers (Puri et al, 2013). Because of severe alterations in TfR and endosomes, we hypothesised that the function of the transmembrane protein ATG9A may be impaired, contributing to the observed autophagy deficits. In control brains, in the CA1 cell body area, ATG9A had a perinuclear localisation, as expected, while in NDR1/2 KOs ATG9A localised to more distal dendritic regions and had a more punctate distribution (Fig 5A). ATG9A mislocalisation was also prominent in upper cortical areas, which are most affected by neurodegeneration (Fig S5A). Interestingly, the levels of ATG9A were increased in the brains of 12-wk-old NDR1/2 KO mice (Fig 5B), even though no change was detected at P20 (Table S1), indicating that there is a chronic accumulation of this autophagy component. We co-stained ATG9A with a well-established Golgi marker, GM130, and found that although in control CA1 neurons the vast majority of ATG9A co-localises with GM130, co-localisation is significantly reduced in NDR1/2 KO neurons (Fig 5C). ATG9A mislocalisation was observed as early as P20 (Fig S5B), supporting the idea that defects in ATG9A and TfR trafficking, downstream of NDR substrates, precede accumulation of p62 aggregates. However, despite having a punctate distribution, similar to TfR accumulations, ATG9A did not co-localise with either TfR or VPS35 (Fig S5C and D).

Strikingly, ATG9A clusters were observed in stratum oriens in NDR1/2 KOs, corresponding to the dendritic arbours of CA1 neurons

---

genotype for p62, and n = 29 control and 24 NDR1/2 KO neurons from two mice per genotype for LC3. **(D)** Immunofluorescence staining of p62 and ubiquitin in the CA1 area at 12 wk. White arrows indicate co-localisation between p62 and ubiquitin puncta. Scale bars: 10 μm. The scatter plot shows quantification of p62 and ubiquitin co-localisation, expressed as a Pearson correlation coefficient. The data were analysed with a Mann–Whitney test. n = 30 measurements from three mice/genotype. **(F, G)** Western blot analyses of LC3 levels in lysates from DIV13 rat primary cortical neurons infected with lentiviruses expressing a scramble (SCRM) shRNA and shRNAs targeting NDR1 and NDR2 (F) or Raph1(G). The cells were treated with DMSO or 100 nM of Bafilomycin A1 (Baf.) for 4 h before lysis. Tubulin was used as a loading control. The bar graphs show quantification of the LC3II bands normalised against the tubulin levels, and the data were analysed using ordinary one-way ANOVAs with Tukey's post hoc test. n = 6 samples/group from three independent experiments, two technical replicates. The scatter plots show quantifications of the absolute increase in LC3II between the DMSO and the bafilomycin A1 conditions. n = 6 measurements/group from three independent experiments, two technical replicates. Source data are available online for this figure.

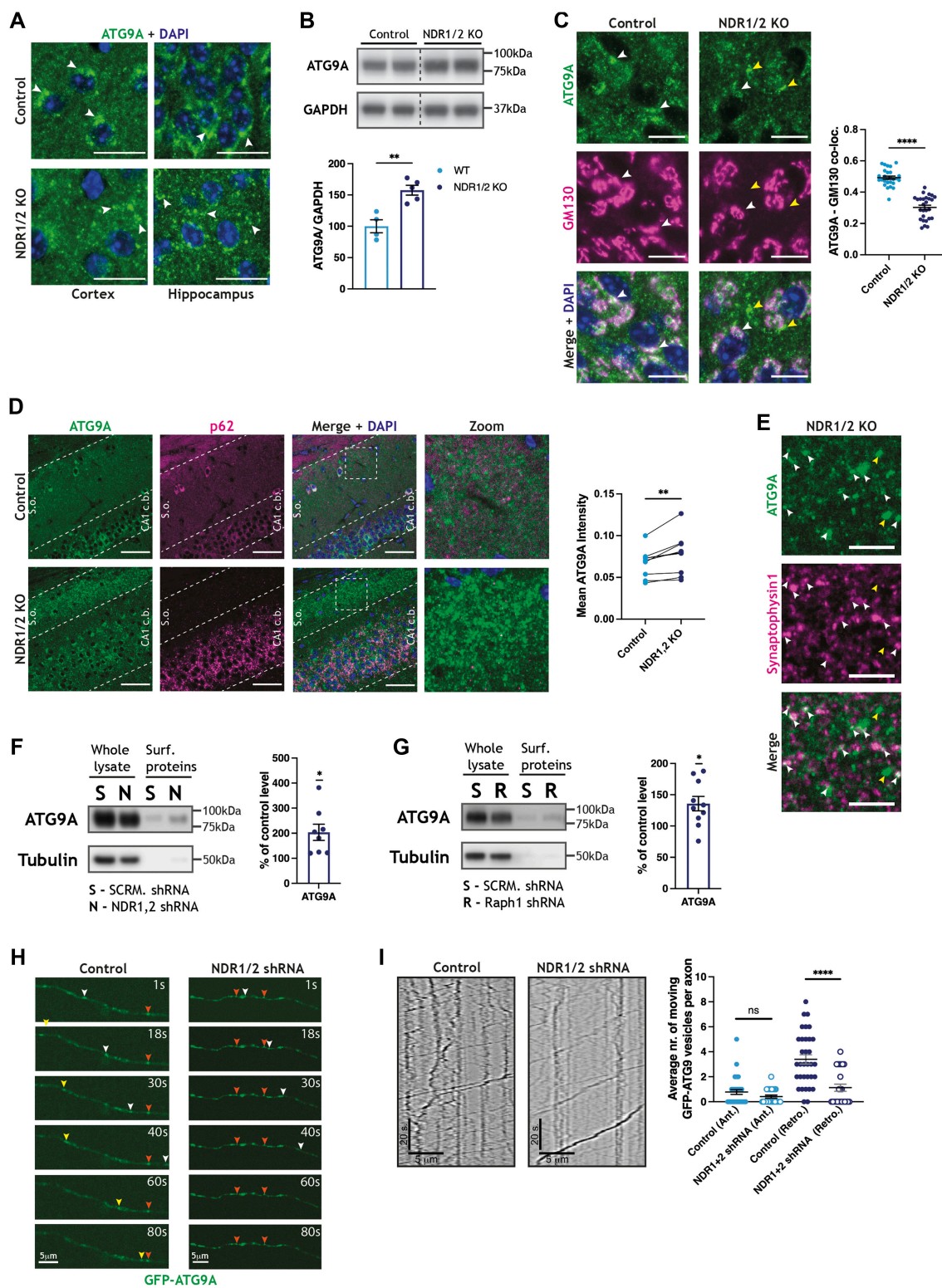

**Figure 5.  ATG9A is mislocalised in NDR1/2 KO brains.**
**(A)** Immunofluorescence staining of ATG9A in brain slices from 12-wk-old NDR1/2 knockout and control mice. White arrows indicate areas with increased endogenous ATG9A. Scale bars: 10 μm. **(B)** Western blot analyses of ATG9A in cortical lysates from 6-wk-old NDR1/2 KO and control mice. GAPDH was used as a loading control. The bar graph shows quantification of ATG9A normalised against GAPDH, and the data were analysed with an unpaired t test. n = 4–5 mice/group. **(C)** Immunofluorescence staining of ATG9A and the Golgi marker GM130 in the CA1 area of 12-wk-old mice. White arrows indicate areas where ATG9A co-localises with GM130. Yellow arrows indicate areas where ATG9A does not co-localise with GM130 in NDR1/2 knockout mice. Scale bars: 10 μm. The scatter plot shows quantification of ATG9A and GM130 co-localisation, expressed as a Pearson correlation coefficient, and the data were analysed with a Mann–Whitney test. n = 28 measurements from three mice/genotype.

synapsing with incoming CA3 axons (Fig 5D and Video 1 and Video 2). To test if ATG9A puncta in stratum oriens are at synapses, we co-stained ATG9A with the presynaptic marker synaptophysin 1. There was a substantial co-localisation between ATG9A and synaptophysin 1 (Fig 5E), indicating that ATG9A accumulations were partially localised near synaptic regions. Because TfR and Chl1 trafficking were altered in NDR1/2 shRNA– or Raph1 shRNA–infected neurons, we decided to assess if ATG9A surface levels can be detected in neurons, and if these are altered when NDR1/2 or Raph1 is depleted. Indeed, surface ATG9A could be detected by surface biotinylation in primary neurons, although this represented only a very small fraction of the total ATG9A (Fig 5F and G). In both NDR1/2 shRNA– and Raph1 shRNA–infected neurons, significantly more ATG9A was present at the surface compared to control neurons, confirming that ATG9A trafficking is altered (Fig 5F and G).

In *C. elegans*, ATG9A endocytosis and recycling at presynaptic boutons are essential for autophagy and neuronal development (Stavoe et al, 2016; Yang et al, 2022). It is also known that autophagosomes formed at axons and presynaptic boutons travel to neuronal cell bodies for lysosomal fusion and degradation (Maday et al, 2012; Maday & Holzbaur, 2016). For this reason, we decided to inspect axonal ATG9A transport using live imaging in primary neurons. We used cultured hippocampus neurons infected with scramble or NDR1 shRNA and NDR2 shRNA–expressing viral vectors and transfected with GFP-ATG9A (Fig 5H and I and Video 3 and Video 4). In a similar fashion to autophagosomes (Maday et al, 2014), most ATG9A vesicles are trafficked retrogradely in axons of both control and NDR1/2-depleted cells (Fig 5H and I and Video 3 and Video 4). However, significantly less ATG9A vesicles were mobile in NDR1/2 shRNA neurons, compared to the scramble shRNA condition, although net displacement and velocity were not changed (Figs 5I and S5E and Video 3 and Video 4). Furthermore, numerous stationary ATG9A clusters were observed in the axons of NDR1/2 shRNA neurons (Fig 5H, orange arrowheads). A reduction in the number of ATG9A-positive vesicles being trafficked in axons could be a result of reduced endocytosis of ATG9A from the plasma membrane, and matches the phenotype of NDR1/2 KO mice, where ATG9A accumulates in the stratum oriens. Our working model proposes that reduced endocytosis of ATG9A could lead to reduced retrograde trafficking in axons and result in the observed drastic mislocalisation of ATG9A from the Golgi to synaptic regions.

Efficient cycling of ATG9A between different membrane compartments is essential for efficient autophagy (Young et al, 2006; Longatti et al, 2012; Longatti & Tooze, 2012; Puri et al, 2013; Popovic & Dikic, 2014; Imai et al, 2016; Ivankovic et al, 2020), so we also assessed the trafficking of transfected RFP-LC3 in the axons of NDR1/2 shRNA neurons. RFP-labelled autophagosomes predominantly travelled in the retrograde direction, as previously reported (Maday et al, 2012). We found a similar reduction in the number of mobile, retrogradely moving autophagosomes, labelled with RFP-LC3, but no changes in velocity or displacement. This indicates that formation or initiation of movement, rather than transport of autophagosomes, was altered (Fig S5F). Therefore, we postulate that the impairment in ATG9A trafficking is a major cause of defective autophagy and consequent neurodegeneration in NDR1/2 KO mice.

## Deleting Ndr1 and Ndr2 in adult mice is sufficient to replicate the phenotype of the conditional knockout mice

Because NEX expression starts at E11.5, before neuronal development is completed, we sought to assess if the changes seen in the brains of NDR1/2 conditional knockout mice were due to a developmental defect. For this, we employed an inducible knockout mouse approach, using NEX-Cre$^{ERT2}$–expressing mice (Agarwal et al, 2012) crossed with *Ndr1*$^{KO}$ and *Ndr2*$^{flox}$ mice, which were injected with tamoxifen twice a day, for 5 d, to induce Cre expression. Identification of recombined neurons is possible due to a reporter gene, Ai14-TdTomato, which switches on as soon as Cre becomes active (Madisen et al, 2010). Tamoxifen administration started after the mice reached adulthood, at 12 wk of age. After tamoxifen treatment, we aged the mice further, until 10–12 mo of age, before analysing the brains.

As previously reported (Agarwal et al, 2012), Cre recombination in the cortex was partial, in a scattered "salt-and-pepper" fashion, as observed from Ai14-TdTomato expression. In particular, recombination was low in the upper cortical layers, and cortical thickness and the expression of GFAP in the cortex were not different between controls and NDR1/2-inducible KOs (iKOs) (Fig 6A and B). By contrast, in the hippocampus CA1-CA3 areas Cre recombination efficiency was close to 100% (Fig 6A), as expected based on previous reports (Agarwal et al, 2012). In NDR1/2 iKOs, the cell body layering of CA1 was disrupted, stratum radiatum was smaller and hippocampal GFAP staining was highly increased (Fig 6A and B), similarly to NEX-Cre conditional NDR1/2 KO mice. These results indicate that deletion of NDR2 in NDR1 knockout adult mice leads to loss of neuropil and degeneration in the hippocampus, while it is possible that the lack of changes in the cortex of NDR1/2 iKO mice is due to the low recombination rate.

Despite a very efficient recombination in the hippocampus, not all the neurons in the CA1 area had reporter expression (Fig 6C). This enabled us to compare side by side neurons that do not express

---

**(D)** Immunofluorescence staining of ATG9A and p62 at 12 wk. White dotted lines delineate the stratum oriens (s.o.) area and the CA1 cell body area (CA1 c.b.). Scale bars: 50 μm. **(E)** Immunofluorescence stainings of ATG9A and the presynaptic marker synaptophysin 1 in stratum oriens at 12 wk. White arrows show ATG9A co-localising with synaptophysin 1, and yellow arrows show ATG9A puncta that do not co-localise with synaptophysin 1. Scale bars: 5 μm. **(F, G)** Western blot analyses of surface biotinylation experiments in DIV12 rat cortical neurons infected with scramble (SCRM) shRNA and NDR1,2 shRNA lentivirus (F) or Raph1 shRNA (G). Surface levels of ATG9A were normalised against input. Bar graphs show surface ATG9A levels that are expressed as a percentage of the corresponding SCRM shRNA control level. The data were obtained in the same experiment shown in Fig 3B and E; therefore, the same tubulin control is used. The data were analysed using a one-sample *t* test. n = 6 samples/condition from three independent experiments. **(H, I)** Representative images (H) and kymographs (I) of ATG9A vesicle movement in the axons of DIV11 cultured rat hippocampal neurons, previously infected with a scramble shRNA lentiviral vector as a control or NDR1 and NDR2 shRNA–expressing lentiviruses. The total number of moving particles were quantified from kymographs. The data were analysed using Mann–Whitney tests. n = 138 mobile particles from 33 cells for scramble shRNA, and n = 37 mobile particles from 24 cells for NDR1/2 shRNA.
Source data are available online for this figure.

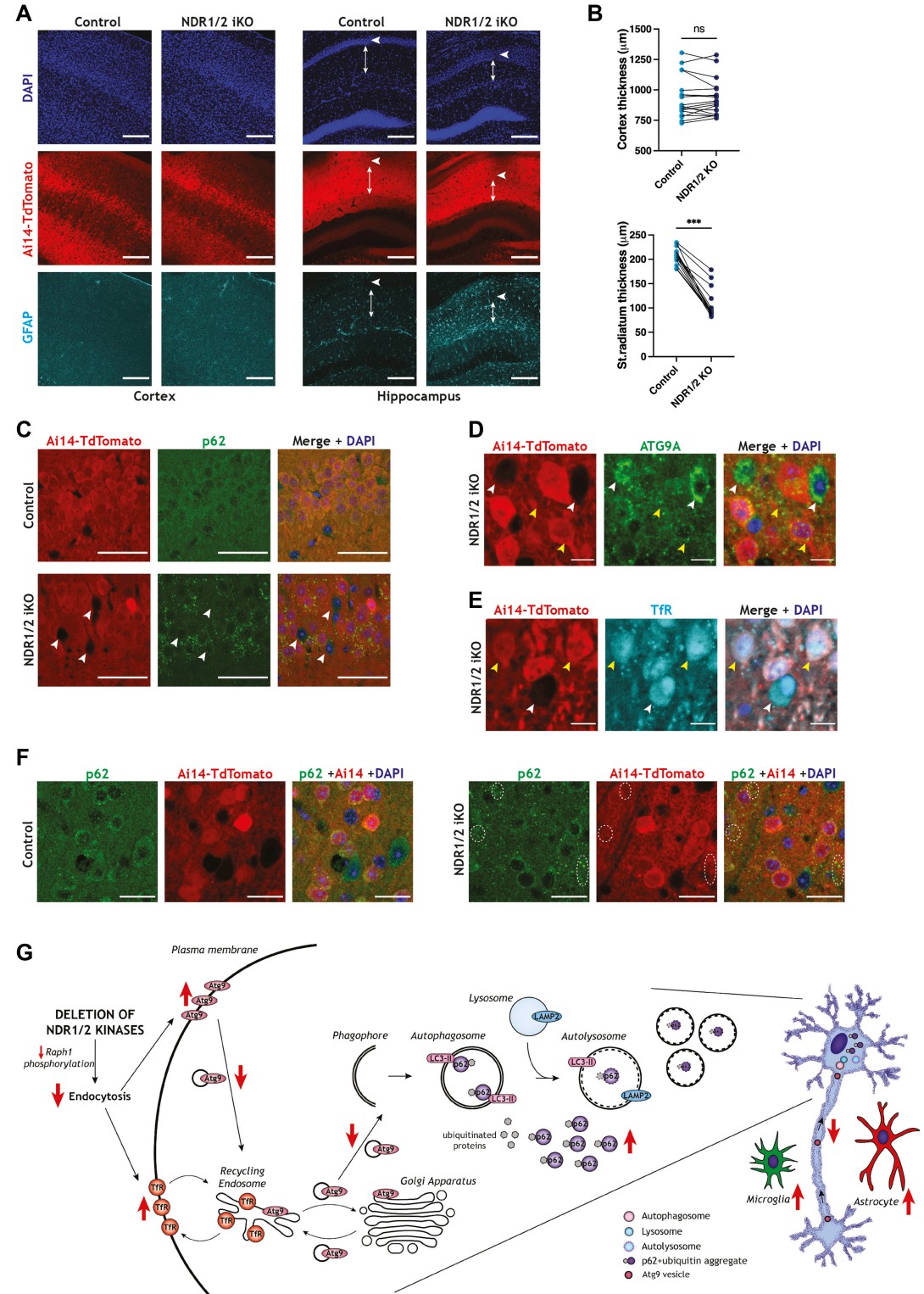

**Figure 6. Dual loss of NDR1/2 in adults is sufficient to induce p62 accumulation, ATG9A mislocalisation and neuropil loss.**
**(A)** Immunofluorescence staining of GFAP in the brains of NDR1/2 inducible knockout (NDR1/2 iKO) and control mice. White arrows show the distribution of neurons within the CA1 cell body layer. Arrowed lines show the length of stratum radiatum. Scale bars: 200 μm. **(B)** Graphs showing quantifications of cortex and stratum radiatum thickness. The data were analysed using paired *t* tests. n = 18 measurements from three mice/genotype. **(C)** Immunofluorescence staining of p62 in NDR1/2 iKO and control mice in the CA1 area. White arrows show Ai14-TdTomato−negative cells, which have not been recombined and lack p62 accumulations, and scale bars represent 50 μm.
**(D, E)** Immunofluorescence staining of ATG9A (D) and transferrin receptor (TfR) (E) in the brains of NDR1/2 iKO and control mice in the CA1 area. White arrows show Ai14-TdTomato−negative cells, which have a perinuclear distribution of ATG9A (D) and lack TfR accumulations (E). Yellow arrows show Ai14-TdTomato−positive cells, which

NDR kinases with neurons expressing only NDR2. Immunofluorescence stainings of p62 revealed p62 accumulations in the CA1 area of NDR1/2 iKO mice in Ai14-TdTomato–positive cells (Fig 6C, yellow arrows), while Ai14-TdTomato–negative neurons lacked such p62 accumulations (Fig 6C, white arrows). Similar results were observed with ATG9A and TfR in CA1 neurons. Ai14-TdTomato–negative cells had a perinuclear ATG9A distribution and lacked TfR puncta (Fig 6D and E, white arrows), as seen in control animals. By contrast, in Ai14-TdTomato–positive neurons, less ATG9A was found in the perinuclear area, while additional ATG9A puncta had a dispersed distribution, and TfR accumulations appeared concentrated in these cells (Fig 6D and E, yellow arrows). Finally, immunofluorescence assessments from areas of the cortex with the highest recombination rate revealed that p62 puncta accumulated in NDR1/2 iKO mice (Fig 6F, white dotted circles), but this was to a lesser extent than in the hippocampus, in accordance with lower recombination levels. In conclusion, these results show that deleting NDR kinases in adult neurons is sufficient to replicate the TfR trafficking deficits, ATG9A mislocalisation and p62 accumulation phenotypes observed in NDR1/2 conditional knockout brains. Importantly, these results indicate that the roles of NDR1/2 in neuronal homeostasis are not due to their function during neuronal development.

Our results are consistent with a model in which NDR1/2 kinases regulate neuronal endocytosis and membrane trafficking via multiple substrates, one of which is the endocytic protein Raph1. Upon loss of NDR1/2, TfR cycling is altered and ATG9A surface levels, neuronal localisation and its axonal transport are changed, highlighting a defect in ATG9A trafficking. We propose that the impairment in ATG9A localisation and trafficking contributes to reduced autophagy and the subsequent accumulation of p62-positive and ubiquitinated proteins that is observed in NDR1/2 KO mice (Fig 6G). It is worth mentioning that other NDR1/2 substrates and downstream signalling events could also impair autophagy and protein homeostasis in parallel.

## Discussion

In this study, we report a novel function of NDR1 and NDR2 in neuronal protein homeostasis and autophagy, which is critical for preserving neuronal health. NDR1 and NDR2 have been implicated in YAP1 regulation and tumorigenesis (Cornils et al, 2010; Zhang et al, 2015). NEX-Cre–mediated deletion of the other two members of the NDR kinase family, LATS1 and LATS2, causes YAP1 activation and tumour formation in brain (Eder et al, 2020). However, we did not find any brain tumours in NDR1/2 KOs. Thus, our study shows that NDR1 and NDR2 do not regulate YAP1 in NEX-positive neural precursors. Instead, NDR1/2 KO brains had significant increases in proteins associated with human neurodegenerative diseases (e.g.,

ApoE, Htt, APP, and PRNP), highlighting similarities between this mouse model and human disorders. *Ndr2* loss-of-function mutations are found in canine early retinal degeneration (Goldstein et al, 2010), and loss of *Ndr1* leads to compromised interneuron numbers in mouse retina. Neurodegeneration was not observed in these *Ndr1* knockouts and this is likely because of the presence of NDR2 (Leger et al, 2018). In our study, we investigate the dual loss of both NDR1 and NDR2 in vivo in mouse neurons, and reveal their compensatory functions in neuronal protein homeostasis, autophagy and neurodegeneration. Our findings indicate that NDR kinases could be novel targets for protection against neurodegeneration.

To gain mechanistic insight into NDR1/2 signalling in vivo, we used quantitative proteomics approaches in double knockout mice. We reveal NDR1/2 kinases as novel regulators of neuronal protein homeostasis. Our TMT-based quantitative proteomics experiments were done at P20, when there was no difference in hippocampal GFAP levels between controls and double KOs, to allow for identification of the earliest changes directly downstream of NDR kinases. The phosphoproteome analysis did not reveal a direct autophagy protein as an NDR1/2 substrate candidate, although PI4KB has been previously implicated in autophagy (Judith et al, 2019). Instead, NDR1/2 phosphorylates a repertoire of membrane trafficking regulators. Using in vitro kinase assays, we validated one of the top candidates, Raph1, as a novel NDR1/2 substrate. The data presented here are of critical value for understanding the neuronal functions and downstream effectors NDR1/2 kinases.

Given the similarities between our NDR1/2 KO model and brain-specific autophagy defective mouse models (Hara et al, 2006; Komatsu et al, 2006), we hypothesised that autophagy may be affected. We show that, together, NDR1 and NDR2 are essential regulators of autophagosome formation in neurons in vivo. The reduced numbers of autophagosomes in neuronal cell bodies and increased overall p62 in NDR1/2 KO brains, as well as reduced LC3 trafficking in axons and reduced LC3 lipidation in primary neurons expressing NDR1/2 shRNA, collectively indicate that both autophagosome formation and degradation could be affected in neurons that lack NDR1 and NDR2. Our results agree with NDR1's reported positive roles in autophagy (Joffre et al, 2015; Martin et al, 2019). Interestingly, while this study was being prepared a recent publication identified the *Drosophila* NDR kinase trc as a binding partner of LC3 (Tsapras et al, 2022). Previously, NDR1 was shown to enhance autophagy by regulating the interaction of Beclin-1 with its effectors (Joffre et al, 2015) and by increasing Exportin-1 activity, leading to increased Beclin-1 levels in the cytoplasm (Martin et al, 2019). In our NDR1/2 KO mouse model, total protein levels and the phosphorylations detected on Exportin-1 or Beclin-1 were not altered (Tables S1 and S5). We also did not observe any changes in mTORC1 downstream effectors or ribosomal proteins, as would be expected from mTORC1-mediated transcriptional regulation, but we cannot rule out changes in gene expression. Although we tried to

---

exhibit a punctate distribution of ATG9A (D) and a high number of TfR puncta (E). Scale bars: 10 μm. **(F)** Immunofluorescence staining of p62 in areas of the cortex with high Ai14-TdTomato expression. White circles highlight areas with p62 accumulations in NDR1/2 iKO brains. Scale bars: 20 μm. **(G)** Schematic diagram depicting membrane trafficking events affected by NDR1/2 in neurons. Loss of NDR1/2 kinases down-regulates endocytosis, likely via a reduction in phosphorylation of Raph1/Lpd. As a result, surface levels of ATG9A and TfR are increased. Altered endocytosis of ATG9A causes reduced axonal retrograde ATG9A transport and reduced cell body/Golgi localisation of ATG9A. ATG9A mislocalisation interferes with autophagosome formation, leading to progressive accumulation of p62-ubiquitinated proteins.

account for the presence of activated glia in KO brains, some neuron-specific proteomics differences could still go undetected.

The autophagy pathway relies heavily on efficient membrane trafficking (Lamb et al, 2013; Puri et al, 2013, 2020; Judith et al, 2019). Considering the roles of NDR1/2 substrates identified in this study and previously (Ultanir et al, 2012), we hypothesised that NDR kinases may play a role in autophagy via their involvement in membrane trafficking and endocytosis. In the list of 11 NDR1/2 substrate candidates, that we identified using with HXRXXS* phosphorylation motif (Fig 2E), the top eight candidates are involved in endomembrane trafficking. Additionally, three of these eight candidates have also been reported as NDR substrates using a chemical genetic substrate identification screen (Ultanir et al, 2012). Our results indicate that by orchestrating multiple substrates, NDR1/2 regulate membrane trafficking. We showed pronounced effects on TfR trafficking, in particular Tf endocytosis, when NDR1/2 or their novel substrate Raph1 are knocked down in neurons. Raph1/Lpd interacts with and functions via the actin regulators Ena/VASP and Scar/Wave-Arp2/3 complexes, to mediate effective cell migration and neuronal morphogenesis (Krause et al, 2004; Michael et al, 2010; Law et al, 2013; Sundararajan et al, 2019). In addition, Raph1/Lpd is required for CME and fast endophilin-mediated endocytosis (Vehlow et al, 2013; Boucrot et al, 2015; Chan Wah Hak et al, 2018), explaining the effect of Raph1 shRNA on Tf uptake. However, it is worth mentioning that this seems to be a neuron-specific function, since in HeLa cells Lpd knockdown impaired receptor-induced CME of EGFR, but not constitutive TfR CME (Vehlow et al, 2013). Interestingly, similarly to NDR kinases, Lpd is also required for neuronal dendritic arborisation (Tasaka et al, 2012), suggesting that a common role in endocytosis and autophagy, as observed here, may contribute to this function.

Considering the strong impact of NDR1/2 loss on membrane trafficking, we next hypothesised that NDR1/2 loss could impact the trafficking and localisation of the only transmembrane autophagy protein, ATG9A. shRNA-mediated depletion of NDR1 and NDR2 resulted in increased cell surface levels of ATG9A, indicating that ATG9A endocytosis could also be affected by NDR1/2, perhaps owing to co-trafficking of ATG9A and TfR (Longatti & Tooze, 2012; Longatti et al, 2012; Puri et al, 2013). ATG9A is known to be endocytosed from the plasma membrane via CME, and when endocytosis is inhibited with Dynasore, dominant-negative dynamin or siRNA targeting AP2, ATG9A disperses from Golgi and accumulates in a different compartment close to the plasma membrane (Puri et al, 2013, 2014). In NDR1/2 KO brains, both TfR and ATG9A formed punctate accumulations that were mislocalised. In *C. elegans* neurons, ATG9A is present at synaptic sites and is essential for autophagy at synapses (Stavoe et al, 2016; Yang et al, 2022). Inhibition of endocytosis in these neurons also caused an accumulation of ATG9A, in this case in discrete locations near the synaptic sites, where it co-localised with clathrin (Yang et al, 2022). In NDR1/2 KO mouse brains, ATG9A severely accumulated in peripheral regions of neurons, where it often co-localised with the synaptic marker synaptophysin 1. In addition, neurons expressing NDR1/2 shRNA exhibited fewer GFP-ATG9A–positive vesicles trafficking retrogradely in axons, in agreement with potentially reduced endocytosis of ATG9A in axon boutons. ATG9A's cycling between endosomal compartments and Golgi is known to be critical for

starvation-induced autophagy in dividing cells and for constitutive autophagy in neurons (Young et al, 2006; Orsi et al, 2012; Puri et al, 2013, 2020; Popovic & Dikic, 2014; Imai et al, 2016; Judith et al, 2019; Ivankovic et al, 2020). Given the clear ATG9A mislocalisation and neurodegeneration phenotypes exhibited by NDR1/2 knockouts, we attribute the autophagy deficit in these mice to impaired ATG9A function. Interestingly, altered ATG9A localisation can be caused by the Parkinson's disease–associated VPS35 mutation D620N, which also impairs autophagy (Zavodszky et al, 2014a, 2014b). Moreover, loss-of-function mutations in another PD-linked gene, encoding parkin, also cause ATG9A mislocalisation downstream of VPS35 (Williams et al, 2018), further highlighting that efficient ATG9A trafficking is important to prevent neurodegeneration.

Additional impairments in the regulation of the endolysosomal system or defects at the plasma membrane could also contribute to the accumulation of ubiquitinated proteins and p62, and to the neurodegeneration phenotype characterised here. Although we found no differences in lysosomal proteins in the proteomics analysis and in stainings (Fig S4E), we cannot rule out a contribution from lysosomes to the observed phenotype. The NDR1/2 knockout/knockdown phenotypes that we demonstrate in this study are expected to be a cumulative effect of misregulation of multiple NDR1/2 substrates. Although we characterise one strand of NDR1/2 signalling via Raph1, deciphering the roles of NDR1/2 and each of its substrates will require additional studies. Further investigations would also be needed to determine upstream activation signals for NDR1/2 kinases. Nonetheless, our findings significantly expand our understanding of the cellular and in vivo roles of NDR1/2 kinases in mammalian neurons.

# Materials and Methods

### Animals

All mouse and rat handling procedures were performed in accordance with the regulations of the Animal (Scientific Procedures) Act 1986. Animal studies and breeding were approved by the Francis Crick Institute ethical committee and performed under UK Home Office project license (P5E6B5A4B). Mice were housed in an animal facility of the Francis Crick Institute on an alternating 12-h light–dark cycle with free access to food and water. All lines are on a C57BL/6J background. Both male and female mice were used and randomly allocated to experimental groups according to genotypes. As much as possible comparisons were made between littermates with different genotypes.

NEX-Cre (*Neurod6*$^{tm1(cre)Kan}$, MGI:2668659) and Nex-CreERT2 (*Neurod6*$^{tm2.1(cre/ERT2)Kan}$) mice were a gift from Dr. Klaus Nave and Dr. Markus Schwab. *Ndr1*$^{KO/KO}$ (B6;129P2-*Stk38*$^{tm1/FMI}$) and *Ndr2*$^{flox/flox}$ (B6CF2;129P2-*Stk38l*$^{tm3/BAH-FMI}$) mice were provided by Dr. Brian Hemming. Ai14 (B6;129S6-*Gt(ROSA)26Sor*$^{tm14(CAG-tdtomato)}$$^{Hze}$/J; Stock No: 007908) and Thy1-YFP lines (B6.Cg-Tg(Thy1-YFP)HJrs/J; Stock No: 003782) were purchased from Jackson Laboratories.

For the *Ndr2*-inducible knockout approach, *Ndr1*$^{KO}$*Ndr2*$^{flox}$ mice were crossed with mice expressing *Nex*$^{CreERT2}$. After the mice reached adulthood, at 12 wk of age, they received a 5-d course of tamoxifen (Sigma-Aldrich) in corn oil—1 mg, twice daily—to induce

Cre expression. After the tamoxifen injections, the mice were weighed weekly and aged further until at least 30 wk of age, before harvesting of the brain.

For Western blot analyses, brains were harvested at P20 or 6 wk of age. The mice were culled by cervical dislocation, the brain was removed from the skull, and the hippocampus or cortex was dissected and flash-frozen in liquid nitrogen. For immunofluorescence, P20-, 12-wk- and 20-wk-old animals were transcardially perfused with 4% ice-cold PFA under terminal anaesthesia before brain harvesting and post-fixation in 4% PFA overnight.

### Immunofluorescence, histology, and imaging

50-$\mu$m-thick coronal slices were obtained from the fixed brains with a Leica VT1000 S vibrating blade microtome (Leica) and used for immunofluorescence staining. The slices were blocked in buffer containing 10% serum and 0.02% Triton X in PBS and then incubated in primary antibodies overnight at 4°C. For the following antibodies, a 20-min antigen retrieval step in citrate buffer (10 mM sodium citrate and 0.05% Tween-20, pH 6.0) at 95°C was carried out before blocking: ubiquitin, TfR, ATG9A, GM130 and VPS35. Secondary antibody incubation was carried out for 1 h at room temperature, and nuclei were stained using DAPI. Sections were mounted on slides with Fluoromount. All images were acquired with a Zeiss Invert 880 confocal microscope. Images obtained using the 4× (Fig 1E) and 10× (Figs 1C, D, and H, S1A, 2A, 4A, and 6A) objectives were acquired as single images, whereas the rest of the images were acquired with the 40× or 63× objective as Z-stacks (1-$\mu$m interval) and three slices were maximally projected to obtain the images used in the figure panels.

For histological analysis, fixed brains were dehydrated and embedded in paraffin, after completion of 4% PFA fixation. 4-$\mu$m sections were prepared and stained with H&E. Images were acquired using Olympus VS120 Slide Scanner.

### Primary neuronal cell culture

Rat cortical and hippocampal neurons were cultured from E16.5 embryos from a WT Long Evans mother. All the hippocampi and all the cortices from one litter were dissected, and all tissues from the same brain area were pooled together. Dissociation of neurons was carried out by incubation with 0.25% trypsin (Gibco) for 20 min at 37°C, followed by Hanks' Balanced Salt Solution (Gibco) washes and resuspension. The cells were then counted and plated on previously coated surfaces in plating media. The following densities were used for plating: 200,000 cells per 18-mm glass coverslip, 300,000 cells per 22-mm well of a 12-well plate, and 500,000 cells per 35-mm glass-bottom dish or per 35-mm plastic dish. The plating media contained 10% FBS, 0.5% dextrose, sodium pyruvate (Gibco), 2 mM glutamine (Gibco) and penicillin/streptomycin in minimum essential medium (Gibco). The coating on the glass coverslips, dishes and plates had been applied overnight at 37°C and contained 60 $\mu$g/ml poly-D-lysine (Sigma-Aldrich) and 2.5 $\mu$g/ml laminin (Sigma-Aldrich) in 0.1 M borate buffer. After ~4 h, the neurons were transferred to maintenance media containing B27 (Sigma-Aldrich), 0.5 mM glutamine, 12.5 $\mu$M glutamate, penicillin/streptomycin and ciprofloxacin in Neurobasal Medium (Gibco). The neurons were kept at 37°C and 5% $CO_2$ until use, and medium change was carried out every 3–4 d, by replacing about 1/3 of the media with fresh maintenance media.

### HEK293T cell culture

HEK293T cells were kept in Dulbecco's modified Eagle medium (Gibco) supplemented with 10% FBS and 1% penicillin/streptomycin. Transfections were carried out using X-tremeGENE 9 (Roche), according to the manufacturer's protocol. For the cells treated with 100 nM Bafilomycin A1 (Sigma-Aldrich) for 4 h, the treatment and subsequent lysis for Western blot analysis happened 48 h after transfection. The HEK293T cells used to purify Raph1 for the kinase assay were lysed 72 h after transfection.

### Western blotting

Flash-frozen mouse brain areas were homogenised by sonication in 2× sample buffer (Pierce LDS Sample Buffer; Thermo Fisher Scientific) containing 0.2 M DTT. Lysates were then centrifuged at 13,000$g$ for 15 min, and supernatants were removed and denatured at 70°C for 10 min. Neurons and HEK293T cells treated with Bafilomycin A1 (Sigma-Aldrich) were lysed directly in 2× sample buffer containing 0.2 M DTT, followed by sonication and incubation at 70°C for 10 min. All lysed samples were run on NuPAGE 4–12% Bis-Tris polyacrylamide gels (Thermo Fisher Scientific) and transferred to a polyvinylidene difluoride membrane (Millipore) using wet transfer. After blocking in 5% non-fat milk in TBST for 1 h, the membranes were incubated with primary antibodies overnight at 4°C or at RT for 1–2h. Peroxidase-conjugated (HRP) secondary antibody incubation was carried out at RT for 1 h. For Western blot detection, the membranes were incubated with ECL (Amersham ECL Prime Western Blotting Detection Reagent) and visualised with a chemiluminescence digital imaging system (Amersham Imager 600 RGB) or developed using film (Amersham Hyperfilm ECL; Western blots in Fig S1B only).

### Protein purification

Constitutively active analogue sensitive NDR1 (NDR1 CA MA) and constitutively active kinase-dead NDR1 (NDR1 CA KD) were purified as detailed in Ultanir et al (2012).

Raph1-WT, Raph1-S192A, Raph1-S571A and Raph1-SA×2 (S192A and S571A) were purified from HEK293T cells using the pCAG-DEST-3×FLAG-2×Strep expression vector. 72 h after transfection, the cells were washed once with cold PBS and lysed in buffer containing 50 mM Tris–HCl, pH 8.0, 5% glycerol, 150 mM NaCl, 10 mM $MgCl_2$, 0.1% Triton X-100, 1× protease inhibitor cocktail (Roche), and 1:5,000 Pierce Universal Nuclease (Thermo Fisher Scientific). The lysates were incubated at 4°C for 30 min, with rotation, to allow for solubilisation, and then centrifuged at 21,000$g$ for 15 min at 4°C. To immunoprecipitate FLAG-tagged proteins, the supernatant was incubated with anti-FLAG M2 Affinity Gel (Sigma-Aldrich) for 1 h at 4°C, with rotation. Proteins bound on FLAG beads were washed once with lysis buffer and once with a wash buffer containing 50 mM Tris–HCl, pH 8.0, 150 mM NaCl, 10mM $MgCl_2$, and 1× protease inhibitor cocktail. Raph1/Lpd is heavily phosphorylated under normal

conditions, so the bead-bound proteins were dephosphorylated in wash buffer containing 1:50 λ-Protein Phosphatase (New England Biolabs) and 1 mM MnCl$_2$ to make the phosphosites more accessible for a kinase assay. The dephosphorylation was carried out at room temperature for 30 min, and the samples were kept rotating. The bead-bound proteins were then loaded onto prewashed Pierce Centrifuge Columns (Thermo Fisher Scientific) and washed twice with high-salt wash buffer (500 mM NaCl) and twice with normal-salt wash buffer. Bound proteins were eluted in wash buffer containing 100 μg/ml 3×FLAG peptide.

### In vitro kinase assays

To establish the concentration of the proteins needed for kinase assays, purified proteins were separated via gel electrophoresis together with known amounts of BSA. The gel was stained with Coomassie stain and imaged using colorimetric assessment on a digital imaging system (Amersham Imager 600RGB). The intensity of the bands was quantified and the concentration of Raph1 and NDR1 constructs established based on the known amounts of BSA loaded onto the same gel.

For the kinase assay, 500 ng Raph1 was incubated with 50 ng NDR1 in buffer containing 20 mM Tris–HCl, pH 7.5, 10 mM MgCl$_2$, l μM okadaic acid, 1 mM DTT, 1× protease inhibitor cocktail (Roche), 100 μM ATP and 0.5 mM 6-benzyl-ATPγS (Biolog) at 30°C, with rotation, for 30 min. The reactions were quenched with 20 mM EDTA and alkylated with 5 mM p-nitrobenzyl mesylate (Abcam) for 30 min at room temperature. The proteins were solubilised in sample buffer (Pierce LDS Sample Buffer; Thermo Fisher Scientific) containing 0.1 M DTT and denatured by incubation at 70°C for 10 min, in preparation for Western blot analysis.

### Tf recycling assay

Rat hippocampal neurons were grown on coverslips and transfected with an empty EGFP-expressing plasmid (pLL3.7) at DIV6. 4 h later, the cells were transduced with lentiviruses expressing shRNAs. At DIV11, the neurons were starved in DMEM (Gibco) with added penicillin/streptomycin for 30 min at 37°C and 5% CO$_2$, and then incubated with 50 μg/ml Tf–Alexa Fluor 568 (Thermo Fisher Scientific) in complete culture media for 20 min at 4°C. Next, the neurons were returned to 37°C and 5% CO$_2$ for 20 min to allow Tf uptake (pulse). At the end of the pulse, the cells were washed three times with PBS and reincubated in culture media devoid of labelled Tf for 0, 20 or 60 min (chase). Following the chase, neurons were washed once more with PBS and fixed with 4% PFA and 4% sucrose in PBS. After mounting of coverslips with Fluoromount, images were acquired with a Zeiss Invert880 confocal microscope.

### Biotinylation of surface proteins

Rat cortex neurons were grown on 35-mm dishes and transduced with lentiviruses expressing shRNAs at DIV6. The protocol for the biotinylation assay was adapted from Twelvetrees et al (2010). At DIV12, the neurons were washed with cold PBS (all wash steps were carried out with PBS supplemented with 1 mM CaCl$_2$ and 0.5 mM MgCl$_2$) and then incubated with 1 mg/ml biotin (EZ-Link Sulfo-NHS-

Biotin; Thermo Fisher Scientific), with shaking, at 4°C for 12 min. The following steps, before cell lysis, were all carried out on ice, using cold solutions. After biotin incubation, the neurons were washed with PBS twice and the biotinylation reaction was quenched with a buffer containing 1 mg/ml BSA in PBS. After three more washes with PBS, the cells were lysed in RIPA buffer (Thermo Fisher Scientific) and pelleted by centrifugation at 13,000g for 15 min in a refrigerated centrifuge (at 4°C). Part of each supernatant was saved as input (total lysate fraction) and mixed with equal volumes of 2× sample buffer, before incubation at 70°C for 10 min, in preparation for Western blot analysis. The rest of the supernatants were incubated with NeutrAvidin beads (Thermo Fisher Scientific) for 2 h at 4°C on a rotator. After three washes with RIPA buffer, the proteins were eluted off the beads in 2× sample buffer and incubated at 70°C for 10 min, in preparation for Western blot analysis.

### Live imaging

Rat hippocampal neurons were grown on 35-mm glass-bottom dishes and transfected with GFP-ATG9A– or RFP-LC3–expressing plasmids at DIV5. 4 h later, the cells were transduced with lentiviruses expressing shRNAs. On DIV11, glass bottoms were placed in the prewarmed chamber (37°C, 5% CO$_2$) of a NIKON CSU-W1 Ti2 Confocal spinning disk microscope with a Dual Prime 95B camera. A 100× (oil) objective was used to identify axons, and axonal imaging was carried out within 200 μM of the soma. 2-min videos were acquired at 2 frames/s. ~25 μm of each imaged axonal segment (15 line width) was selected to generate kymographs using the KymographClear plugin (Mangeol et al, 2016) in FIJI. The kymographs were then filtered (DoG filter with a pixel radius ratio of 2:0.5, signal to noise) and resized so that they all had the same size. Tracing and motility analyses were carried out with the KymoAnalyzer plugin (Neumann et al, 2017) in FIJI. In terms of tracing, only the objects that were present in most of the frames were manually traced. For the generation of displacement and velocity (average) data, the following conversion values were used: pixel size - 0.11 μm; and frames per second - 2.

### Antibodies, plasmids and viral vectors

The following primary antibodies were used for immunofluorescence: mouse GFAP (1:2,000, G6171; Sigma-Aldrich), rat Ctip2 (1:500, ab18465; Abcam), rabbit Iba1 (1: 200, 019-19741; Wako Chemicals), guinea-pig p62 (1:400, GP62-C; Progen), rabbit LC3-B (1:200, 3868; Cell Signaling Technology), rabbit ubiquitin (1:100, ab134953; Abcam), rat LAMP2 (1:300, ab134953; Abcam), mouse transferrin receptor—TfR (1:250, 13-6800; Thermo Fisher Scientific), rabbit ATG9A (1:200, ab108338; Abcam), mouse GM130 (1:200, 610822; BD), rabbit VPS35 (1:200, ab157220; Abcam), mouse synaptophysin 1 (1:500, 101 011; Synaptic Systems) and goat TdTomato (1:400, AB8181-200; SICGEN). All fluorescent secondary antibodies were purchased from Jackson ImmunoResearch and used at 1:500 dilution.

The above-mentioned markers GFAP, p62, ubiquitin, TfR and ATG9A were also used for Western blotting at 1:1,000 dilution, in addition to the following antibodies: mouse GAPDH (1:10,000, ab8245; Abcam), mouse α-tubulin (1:20,000, T9026; Sigma-Aldrich), mouse NDR1 (1:1,000, SAB1408832; Sigma-Aldrich), rabbit NDR2 (1:500, previously published

in Ultanir et al [2012]), mouse PSD95 (1:1,000, MA1-045; Thermo Fisher Scientific), mouse Homer1 (1:1,000, 160011; Synaptic Systems), mouse GluR1 (1:1,000, MAB2263; Millipore), rabbit LC3-B (1:2,000, ab48394; Abcam), rabbit Chl1 (1:1,000, ab106269; Abcam), rabbit Raph1 (1:1,000, ab121619; Abcam) and rabbit thiophosphate ester (1:200, ab92570; Abcam). The Raph1 phospho-S192 antibody is a custom-made rabbit polyclonal antibody raised against the following Raph1 peptide: C-TNQHRRTAS*AGTVScoNH$_2$ (Covalab). All HRP-conjugated secondary antibodies were purchased from Jackson ImmunoResearch and used at 1:10,000 dilution.

The following plasmids were used for transfection: NDR1-CA-HA, NDR1-KD-HA, and NDR1-WT-HA (previously published in Ultanir et al [2012]); GFP-ATG9A (gift from Sharon Tooze); RFP-LC3 (gift from Max Gutierrez); and Raph1-WT-GFP and Raph1-S192A-GFP (generated via site-directed mutagenesis using the Raph1-WT-GFP plasmid). The pCAG-DEST-Raph1-WT-3×FLAG-2×Strep was generated via Gateway cloning according to the manufacturer's protocol (Thermo Fisher Scientific) starting from a pENTR3C-Raph1-WT plasmid and a pCAG-DEST-3×FLAG-2×Strep, which was generated by HIFI assembly (New England Biolabs) using pCAG-EGFP (89684; Addgene) and a geneblock (IDT) harbouring a TEV site, a 3×FLAG, a 6× glycine linker and a Twin-Strep-tag. S192, S571 or both sites were then mutated to alanines using site-directed mutagenesis to obtain phosphomutant Raph1 constructs.

The shRNA-expressing viral vectors (pLV[shRNA]-TagBFP2-U6) were acquired from VectorBuilder. NDR2 shRNA (rStk38l [shRNA#2]), Raph1 shRNA (rRaph1[shRNA#2]) and scramble shRNA (VB190306-1051gge) were ordered from the VectorBuilder database, and the sequence for NDR1 shRNA has been previously published in Ultanir et al (2012).

## Image analysis

Image analysis was carried out using CellProfiler.

The thickness of the cortex, the thickness of the hippocampus and the length of stratum radiatum were measured in DAPI-stained slices from three different mice/genotype. The length of apical dendrites from individual CA1 neurons was measured in Thy1-YFP–positive cells using the YFP signal. The slices were matched between controls and double knockout animals in terms of the relative area of the brain that they belonged to. A straight line was drawn to measure the desired length, and the same angle was kept for this line between matching slices.

For the measurement of Tf-568 signal in the Tf recycling assay, one image/neuron was obtained using the 63× objective and 2× zoom, by opening the pinhole and establishing optimal focus. The EGFP cell fill was used to delineate the cell body area, and Tf signal was measured only in the cell body and normalised against the total surface area. Alternatively, where EGFP cell fill was not available, perinuclear Tf-568 was measured per each neuron.

For the measurement of p62 and LC3 puncta from brain slices, Z-stacks were acquired with the 63× objective, 2× zoom and 1-$\mu$m interval between slices, to cover the entire volume of neurons. YFP and DAPI signals were used to delineate the cytoplasmic area, and the number of puncta was measured only in the cytoplasm. For each neuron, the number of puncta from each slice was normalised against the surface area of the cytoplasm and then added together to obtain the total puncta/cell. The number of TfR puncta was assessed in similarly

acquired Z-stacks from mice that did not express YFP in neurons. The TfR antibody requires an antigen retrieval step that markedly diminishes the YFP signal, making it impossible to use it in quantifications. DAPI staining was used to discard the nuclear area and measure levels of TfR puncta in cytoplasmic areas across the entire slice.

Co-localisation assessments between p62 and ubiquitin, ATG9A and GM130, ATG9A and TfR, and TfR and VPS35 were carried out using Z-stacks acquired with the 63× objective, 3× zoom and 2-$\mu$m interval between slices. The DAPI signal was used to identify nuclei, and the nuclear areas were discounted when assessing the intensity of the above-mentioned markers. Co-localisation between the markers was measured using the Pearson correlation coefficient or the adjusted Rand index.

ATG9A signal in the stratum oriens area was measured in Z-stacks acquired with the 40× objective, to have the entire width of the stratum oriens area within the frame (similar to images in Fig 5D). The Z-stacks, comprising 10 slices acquired with a 1-$\mu$m interval, were projected as an average, and the intensity of the ATG9A signal was measured from the average projection.

## TEM

For TEM, mouse brains were perfusion-fixed in 4% PFA and 50-$\mu$m sections were obtained by vibratome sectioning. The slices to be analysed were stored in 0.1 M PB until further processing. The slices were then transferred to polypropylene 24-well plates (Caplugs Evergreen) and processed using a PELCO BioWave Pro+ microwave (Ted Pella Inc.) and following a protocol adapted from the National Centre for Microscopy and Imaging Research protocol (Deerinck et al, 2010), which was detailed in Eder et al (2020). After processing and embedding of slices into Durcupan ACM resin (Sigma-Aldrich), the blocks were trimmed to a trapezoid around the hippocampus CA1 area. The samples were then sectioned using a UC7 ultramicrotome (Leica Microsystems), and 70-nm sections were picked up on Formvar-coated G50HEX copper grids (Gilder Grids Ltd.). The sections were viewed using a 120-kV Tecnai G2 Spirit TEM (FEI Company), and images were captured using an Orius CCD camera (Gatan Inc.). 22 control cells and 16 NDR1/2 knockout cells from respectively six and three different areas of the CA1 region were imaged at various magnifications. 16,500× images were randomised and anonymised using Advanced Renamer, for unbiased analysis of the mitochondrial morphology.

## Mass spectrometry and data processing

Liquid chromatography–tandem mass spectrometry (LC-MS/MS) was used to analyse differences in the total proteome and the phosphoproteome of P20 NDR1/2 KO mice compared with control animals. The hippocampus was dissected from five control mice and five NDR1/2 KO mice and snap-frozen in liquid nitrogen before lysis and labelling with a TMT 10-plex reagent. Sample preparation, data acquisition and mass spectrometry analysis were carried out as previously described in Eder et al (2020), with some modifications. Briefly, pooled TMT-labelled (Lot# UK288606) samples were cleaned up using a C$_{18}$ Sep Pak Vac 1cc, 50 mg (Waters), and dried. The peptide mixture was then subjected to high-select sequential enrichment of metal oxide affinity chromatography (SMOAC) to capture phosphopeptides. It was first passed through a high-select

TiO$_2$ phospho-enrichment column (A32993; Thermo Fisher Scientific), following the manufacturer's protocol. Flow-through and wash fractions were combined, dried and subsequently used for Fe-NTA phospho-enrichment (A32992; Thermo Fisher Scientific). One-tenth of the combined flow-through and wash fractions from this enrichment was used for total proteome analysis. The eluates from SMOAC were freeze-dried, solubilised and pooled together. Total proteome and phosphoproteome samples were subjected to high-pH reversed-phase fractionation (84868; Thermo Fisher Scientific), dried and resolubilised in 0.1% TFA, prior to LC-MS/MS. The samples were then subjected to HCD MS2 and MSA SPS MS3 fragmentation methods as described in Jiang et al (2017). Data were acquired in data-dependent acquisition mode using an Orbitrap Eclipse Tribrid mass spectrometer (Thermo Fisher Scientific). Acquired data were processed using MaxQuant v 2.0.3.0. Processed data were then analysed using an R-coding script tailored to isobaric labelling mass spectrometry. The script was generated as a hybrid, using the backbone and differential gene expression analysis of the ProteoViz package (Storey et al, 2020) as a general script workflow, and borrowing the normalisation script from the Proteus package (Gierlinski et al, 2018 *Preprint*). Briefly, the "proteinGroups.txt" and "Phospho (STY)Sites.txt" tables were read into matrices and filtered for "reverse" hits, "potential contaminant" and proteins "only identified by site." Data were then normalised using CONSTANd (Maes et al, 2016), log$_2$-transformed and differentially analysed using linear models for microarray data. Significant hits were called proteins and phosphopeptides with $P < 0.005$. Volcano plots were generated using ggrepel (a ggplot2 extension) as part of the tidyverse. Scripts are provided in Supplemental Data 1. The raw data are available via ProteomeXchange with identifier PXD035566.

### Statistical analysis

Statistical analysis was performed using the GraphPad Prism 7 software with the statistical tests indicated in the figure legends. Statistical significance was observed as follows: ns, not significant ($P > 0.05$); *, $P < 0.05$; **, $P < 0.01$; ***, $P < 0.001$; and ****, $P < 0.0001$. All error bars represent SEM.

Where not quantified, Western blots shown are representative of experiments using at least three mice/genotype or repeated in three independent cellular culture experiments. All immunostaining experiments have been repeated in at least three mice/genotype, and representative images have been chosen for figures.

# Supplementary Information

# Acknowledgements

We thank all members of the Ultanir laboratory for valuable discussions and technical help. We thank Dr Alistair Reith and Dr Maximiliano Gutierrez for input and guidance. We thank Emma Nye for providing expertise in histopathology. This research was funded by the Francis Crick Institute, which receives its core funding from the Cancer Research UK (CC2037, CC2134), the UK Medical Research Council (CC2037, CC2134), and the Wellcome Trust (CC2037, CC2134). F Roşianu was funded by a BBSRC-GSK CASE PhD studentship. S Jalal was supported by a Malaysian Public Service Department (PSD) studentship. M Krause acknowledges grants from the Biotechnology and Biological Science Research Council, UK (BB/N000226/1; BB/R015953/1).

## Author Contributions

F Roşianu: conceptualisation, data curation, formal analysis, validation, investigation, methodology, and writing—original draft, review, and editing.
SR Mihaylov: data curation, formal analysis, and investigation.
N Eder: data curation, formal analysis, and investigation.
A Martiniuc: data curation, formal analysis, and writing—original draft.
S Claxton: data curation, investigation, and methodology.
HR Flynn: formal analysis, supervision, investigation, methodology, and project administration.
S Jalal: data curation, formal analysis, and methodology.
M-C Domart: data curation, formal analysis, investigation, and writing—original draft.
L Collinson: formal analysis, supervision, funding acquisition, investigation, methodology, project administration, and writing—original draft.
M Skehel: formal analysis, supervision, funding acquisition, investigation, methodology, and writing—original draft.
AP Snijders: formal analysis, supervision, funding acquisition, investigation, methodology, and writing—original draft.
M Krause: supervision, methodology, and writing—original draft.
SA Tooze: supervision, methodology, and writing—original draft.
SK Ultanir: conceptualisation, formal analysis, funding acquisition, investigation, project administration, and writing—original draft, review, and editing.

## Conflict of Interest Statement

The authors declare that they have no conflict of interest.

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
