## [Reviewer comments · Life Science Alliance]

Life Science Alliance

Loss of NDR1/2 kinases impairs endomembrane trafficking and autophagy leading to neurodegeneration

Flavia Rosianu, Simeon Mihaylov, Noreen Eder, Antonie Martiniuc, Suzanne Claxton, Helen Flynn, Shamsinar Jalal, Marie-Charlotte Domart, Lucy Collinson, Mark Skehel, Ambrosius P. Snijders, Matthias Krause, Sharon Tooze, and Sila Ultanir

DOI: <https://doi.org/10.26508/lsa.202201712>

Corresponding author(s): Sila Ultanir, The Francis Crick Institute

Review Timeline:

Submission Date:	2022-09-06
Editorial Decision:	2022-09-22
Revision Received:	2022-10-17
Editorial Decision:	2022-10-19
Revision Received:	2022-10-28
Accepted:	2022-11-01

Transaction Report:

Please note that the manuscript was previously reviewed at another journal and the reports were taken into account in the decision-making process at Life Science Alliance. Since the original reviews are not subject to Life Science Alliance's transparent review process policy, the reports and author response cannot be published.

September 22, 2022

Re: Life Science Alliance manuscript #LSA-2022-01712-T

Dr. Sila K Ultanir
The Francis Crick Institute
1 Midland Road
London, London NW1 1AT
United Kingdom

Dear Dr. Ultanir,

Thank you for submitting your manuscript entitled "Loss of NDR1/2 kinases impairs endomembrane trafficking and autophagy leading to neurodegeneration" to Life Science Alliance. We invite you to re-submit the manuscript, revised according to your most recent Response to Reviewers.

The typical timeframe for revisions is two months.

When submitting the revision, please include a letter addressing the reviewers' comments point by point. You can use the same Response you just provided, unless there are changes once experiments are completed.

Thank you for this interesting contribution to Life Science Alliance. We are looking forward to receiving your revised manuscript.

Sincerely,

B. MANUSCRIPT ORGANIZATION AND FORMATTING:

October 19, 2022

RE: Life Science Alliance Manuscript #LSA-2022-01712-TR

Dr. Sila K Ultanir
The Francis Crick Institute
1 Midland Road
London NW1 1AT
United Kingdom

Dear Dr. Ultanir,

Thank you for submitting your revised manuscript entitled "Loss of NDR1/2 kinases impairs endomembrane trafficking and autophagy leading to neurodegeneration". We would be happy to publish your paper in Life Science Alliance pending final revisions necessary to meet our formatting guidelines.

- please add the Twitter handle of your host institute/organization as well as your own or/and one of the authors in our system
- please use the [10 author names, et al.] format in your references (i.e. limit the author names to the first 10)
- we encourage you to introduce the panels in your figure legends in alphabetical order
- please upload your video files as separate files
- please double-check your callouts for Figure 3; it appears that panel 3F is missing from your figure and legend
- the Supplemental Data files should be relabeled as Supplemental Tables. Please upload these tables as separate files and update the callouts and legends for the tables in the text
- under the Mouse Handling section, please indicate approval for the animal work, and who provided this approval

Figure Check:

- the Figure 3B and E Tubulin blots appear to be duplicated in Figure 5F and G. If this is accurate, and they are derived from the same experiment, then please mention this in the Figure 5 legend.
- same comment about Figure 4G Tubulin being duplicated in Figure S3D

A. FINAL FILES:

B. MANUSCRIPT ORGANIZATION AND FORMATTING:

Sincerely,

November 1, 2022

RE: Life Science Alliance Manuscript #LSA-2022-01712-TRR

Dr. Sila K Ultanir
The Francis Crick Institute
1 Midland Road
London, London NW1 1AT
United Kingdom

Dear Dr. Ultanir,

Thank you for submitting your Research Article entitled "Loss of NDR1/2 kinases impairs endomembrane trafficking and autophagy leading to neurodegeneration". It is a pleasure to let you know that your manuscript is now accepted for publication in Life Science Alliance. Congratulations on this interesting work.

DISTRIBUTION OF MATERIALS:

Again, congratulations on a very nice paper. I hope you found the review process to be constructive and are pleased with how the manuscript was handled editorially. We look forward to future exciting submissions from your lab.

Sincerely,
